# Current Trends in Nanomaterials for Metal Oxide-Based Conductometric Gas Sensors: Advantages and Limitations. Part 1: 1D and 2D Nanostructures

**DOI:** 10.3390/nano10071392

**Published:** 2020-07-17

**Authors:** Ghenadii Korotcenkov

**Affiliations:** Department of Theoretical Physics, Moldova State University, MD-2009 Chisinau, Moldova; ghkoro@yahoo.com

**Keywords:** nanowires, nanobelts, nanosheets, individual 1D structures, nanowire array, synthesis, fabrication, stability, improvement

## Abstract

This article discusses the main uses of 1D and 2D nanomaterials in the development of conductometric gas sensors based on metal oxides. It is shown that, along with the advantages of these materials, which can improve the parameters of gas sensors, there are a number of disadvantages that significantly limit their use in the development of devices designed for the sensor market.

## 1. Introduction

In the literature, one can find a large number of works aimed at promoting the advantages of nanomaterials for various applications [1,2,3,4,5,6,7,8,9,10,11,12,13,14,15,16,17,18,19,20,21]. As a rule, these articles promote the idea that the use of nanomaterials such as 1D and 2D nanomaterials, as well as hierarchical structures will solve the main problems of devices made on the basis of conventional materials. In this review, we will not follow this tradition. We will try to give a realistic look at the problem of using nanomaterials and show that, along with obvious advantages, 1D and 2D nanomaterials have disadvantages, which in some cases can significantly limit their application.

As an object for consideration, we chose the metal oxide-based gas, vapor and humidity sensors of conductometric type [15,22,23,24,25,26,27,28]. First, metal oxide conductometric sensors are simple, compact, and durable devices that allow controlling the presence in the atmosphere of various toxic and explosive gases and vapors [27,28,29,30,31,32,33]. These sensors are easy-to-manufacture and characterized by low production costs. This means that metal oxide conductometric gas sensors can be mass-produced at a reasonable cost. The principles of their functioning, based on adsorption-desorption processes and catalytic reactions, taking place on the surface of metal oxides and accompanied by an electronic exchange between adsorbed particles and the bulk of metal oxide, are well known. As a result of these reactions, a decrease or increase in the resistance (conductivity) of the gas-sensitive layer occurs, which is proportional to the concentration of the test gas [22,23,24,25,26,27,34]. The processes occurring on the surface of metal oxides during interaction with gases and vapors are repeatedly and in detail described in reviews and books [27,28,35,36,37,38,39,40,41,42,43,44,45,46,47,48,49,50,51,52,53,54,55,56,57,58,59,60,61]. Second, metal oxide conductometric gas sensors from the very beginning of their development were based on the use of nanomaterials. Metal oxide crystallites forming a gas sensitive layer always have a size of less than 100 nm [29,62]. Third, the crystallite size reduction is one of the most common approaches used to improve gas sensor parameters [62,63,64].

As for the metal oxides themselves suitable for gas sensors design, the experiment showed that a variety of materials such as binary oxides (SnO_2_, In_2_O_3_, WO_3_, TiO_2_, Cu_2_O, Fe_2_O_3_, V_2_O_5_, etc.), complex metal oxides (CdIn_2_O_4_, NiTa_2_O_6_, CoTa_2_O_6_, CuTa_2_O_6_, BaSnO_3_, LnFeO_3_, etc.), and nanocomposites (SnO_2_-Fe_2_O_3_, SnO_2_-WO_3_, In_2_O_3_-Fe_2_O_3_, etc.) can be used for these purposes [25,60,65,66,67,68,69,70,71]. At that point, it was established that we do not have an ideal gas-sensitive material [65,72,73]. All the materials have their advantages and disadvantages. Some have low selectivity, others have increased sensitivity to humidity, some are stable only at low temperatures, some require elevated temperatures for efficient operation, etc. Therefore, when choosing a metal oxide-based gas sensing material, it is necessary to take into account the type of sensor being developed, the nature of the gas tested, the technologies used in the manufacture of the sensor, and operating conditions [65,74,75,76].

It should also be borne in mind that metal oxides suitable for the manufacture of gas sensors can form a wide variety of metal oxide nanostructures, which also have their own pronounced features. These are 0D, 1D, 2D, and 3D structures, which include nanodots, nanorods, nanowires, nanotubes, nanobelts, nanosheets, nanofibers, and various hierarchical structures such as nanospheres, nanowhiskers, nanorings, nanoflowers, core-shells, etc. [3,9,67,71,77,78,79,80,81,82,83,84,85,86]. SEM images of some nanostructures are shown in Figure 1.

It is seen that the indicated 0D, 1D, 2D, and 3D nanomaterials due to the structural features will form gas-sensitive layers, which can radically differ from each other even when using the same metal oxide. This means that the choice of a nanomaterial with the necessary structure can become one of the additional methods for optimizing parameters of gas sensors, which forces us to consider in more detail the specifics of gas sensors based on these 0D, 1D, 2D, and 3D nanomaterials. In this review, however, we will not consider 0D structures or nanodots, since almost all nanocrystallites used in the development of conventional gas sensors fall under the definition of nanodots. The properties of these materials and devices based on them are examined in detail and systematized in many reviews and books [23,24,65,66,67,74,94]. In particular, it was shown that, in order to achieve optimal sensor parameters, the sensitive material must have a certain set of properties [26,27,28,65,67,76,94,95,96,97]. First, the sensitive material must have a certain combination of electrophysical, chemical, and catalytic properties [38,65]. Second, for gas sensors to be commercially viable, the gas-sensitive material must be stable during the operation regardless of the surrounding atmosphere and give a reproducible response to the test gas for a prolonged period of time [29,65,67]. Third, to achieve high sensitivity the gas sensitive material must be porous with high gas permeability and a large active surface area. Generally, the surface area of nanomaterials increases as the nanoparticle size decreases. In addition, the crystallite size should be comparable with the Debye length [34]. Therefore, crystallites forming a gas-sensitive layer should have a size less than 20–30 nm and they should not be strongly agglomerated [74]. Otherwise, “dead” or closed zones inaccessible to the gas are formed [74,95]. Fourth, for a fast response and small hysteresis, the pore diameter should be maximal.

The mentioned above testifies that for conventional gas sensors designed on the base of polycrystalline metal oxides, there are quite specific requirements for parameters of gas-sensitive materials. However, it must be borne in mind that the simultaneous fulfillment of all these requirements is not necessary to achieve the desired result. For example, when developing humidity sensors, special attention should be paid to the formation of a material with a certain distribution of pore diameter [28,97]. Too small a pore diameter gives high sensitivity, but in this case the sensor response is slow and it is characterized by large hysteresis, and a narrow range of measured humidity. If the pores of a large diameter predominate in metal oxides, the response will be fast with a small hysteresis, but the sensors can only work in the range of very high humidity, without the ability to detect low humidity levels [98]. A sufficiently detailed description of other approaches that can be used to optimize the parameters of conventional metal oxide-based gas sensors is given in [65,99,100,101,102,103,104,105].

Based on the above, in this review we will focus only on 1D, 2D, and 3D nanostructures. In this review, we will also not analyze the approaches used for functionalizing these nanomaterials in order to improve their gas-sensitive characteristics such as selectivity and sensitivity [106,107,108,109,110,111,112]. They are no different from the approaches used in the manufacture of conventional metal oxide-based gas sensors and based on (i) functionalization with noble-metal catalysts, (ii) tuning the operation temperature, (iii) using heterojunctions, (iv) forming nanocomposites, (v) light activation, and (vi) using filters of various types. A description of these approaches can be found in [99,101,113,114,115,116,117,118,119,120,121,122,123].

Taking into account the wide variety of nanomaterials and the large amount of information obtained during the development of gas sensors based on them, our review will be divided into two parts. The first part, i.e., this article will be devoted to the consideration of 1D and 2D nanomaterials, and the second part will be devoted to the consideration of mesaporous-macroporous materials, hierarchical structures, and nanofibers.

## 2. 1D Structures

### 2.1. 1D Metal Oxide Nanomaterials and Their Synthesis

One-dimensional metal oxide nanostructures, such as nanowires, nanotubes, and nanobelts most often attract the attention of gas sensor developers [81,86,124,125,126,127]. At present, the use of individual 1D nanostructures for gas sensor design is considered as one of the most promising approaches to develop a new generation of gas sensors with an improved stability and reduced temporal drift of the parameters in comparison with conventional devices [128,129].

There are many different techniques that can be used to prepare 1D metal oxide nanostructures, but essentially, they can be grouped into two broad classes (see Figure 2): “Top-down” and “bottom-up” approaches [6,7,130]. The basic idea of the “top-down” approach is to use existing technology developed in the semiconductor industry. This class of techniques uses deposition, etching, and ion beam milling on planar substrates in order to reduce the dimensions of the structures to a nanometer size. The electron beam, focused ion beam, X-ray lithography, nano-imprinting, and scanning probe microscopy techniques are used for the selective removal processes. When using these techniques, it is possible to obtain highly ordered nanostructures [131,132,133], but they are very expensive in terms of both costs and preparation times.

The second approach is the “bottom-up”, which consists of the assembly of molecular building blocks. Among many strategies for the synthesis of oxide 1D structures, a bottom-up growth from the vapor phase is traditionally considered the most cost-effective method of producing 1D structures in large quantities [7,84,130,134,135]. In many cases, this method is called the Vapor-Liquid-Solid (VLS) method, because the growth of 1D structures usually occurs by the vapor-liquid-solid mechanism [136,137]. The growth process is typically accomplished in a low pressure and high temperature furnace. The temperature near the source is elevated sufficiently to melt the source material so it can evaporate. A carrier gas flow brings the vapor to the substrate where nanowires grow with the assistance of catalysts. The catalyst material can be pre-deposited to the substrate for growing or may spontaneously form during the VLS growth process. Therefore, this process can be classified as a metal catalyst or non-metal catalyst VLS. In the metal catalyst VLS method, metals such as Au, Fe, Co, and Ni are used as catalysts. Metal catalysts can be mixed with the source material or spread on the substrate where nanowires grow. Metal catalysts play a crucial role in the synthesis procedure, acting as nucleation sites and promoting the growth of metal oxide nanowires. It is interesting, that during this process, the catalyst particle tends to remain at the tip of the growing nanowire. During this process, the size and density of nanowires on the substrate, used for their growing, can be controlled via changing the dimension and the density of the metal catalyst seeds.

The Vapor-Solid (VS) growth mechanism is different from the VLS growth mechanism, since no catalytic liquid metal is required to promote the growth of 1D metal oxide nanostructures. [138]. Only vapor and solid phases are involved in the growth of 1D nanostructures. Self-catalysis is one of the main factors of this growth. Since there are no catalytic liquid metals, the crystalline solids gain material directly from the vapor without the intermediate liquid phase. As a consequence of the VS growth mechanism, materials anisotropically grow along the preferential axis forming a 1D morphology. One of the main disadvantages of this method is the poor growth control and insufficient ability to integrate 1D nanostructures into fully functional devices. In addition, the high temperature, required for growth, may be a limitation for process substrates or wafers with metals already on top. The advantages of this method include the low cost of the experimental setup, the reduced diameter of the 1D structures, and the high purity of the materials.

Various approaches can be used for the controlled formation of metal catalyst seeds on the substrate surface. [130,135]. Two of them are shown in Figure 2b and Figure 3. Both methods allow controlling the size of metal seeds, and, therefore, controlling the diameter of the nanowires. In the first case (Figure 2), this is achieved through a change in the diameter of polystyrene spheres, and in the second (Figure 3) through a change in the pore diameter in the alumina template [135]. Vertically aligned arrays of carbon nanoposts obtained from nanoporous alumina templates, are used as the nanoscales tamps for printing catalyst nanoparticles.

In contrast to the vapor-gas-phase growth, another main stream of the 1D system synthesis is a solution-phase growth. Growth in the solution phase is a simple and cost-effective way to produce large quantities of 1D nanostructures. There are many different approaches to the production of 1D materials in liquid chemicals, including the polyol method [139], the surfactant method [140], and the hydrothermal/solvothermal technique [141]. Among these methods, the hydrothermal technique, usually carried out in an autoclave, is one of the most versatile and conventional methods, realized in industrial applications because of its high yield, low cost, and moderate-temperature processes. This technology requires simply precursors, mixed with the solution and heated to the required temperature in the autoclave. The desired 1D materials are then obtained as a result of chemical reactions. The disadvantage of this hydrothermal approach is its long synthesis time, which usually takes several hours.

At present, a variety of 1D nanostructures, such as ZnO, In_2_O_3_, CuO, Ga_2_O_3_, MnO_2_, CeO_2_, TiO_2_, and WO_3_, among others, have been explored [88,142,143,144,145,146]. However, the largest number of articles is devoted to the consideration of SnO_2_, In_2_O_3_, and especially ZnO 1D structures. Interest in the 1D ZnO structures is stimulated by the easy synthesis of high-quality and single crystalline 1D ZnO nanostructures [147]. It has been reported that the synthesis of 1D nanostructures based on other gas sensitive metal oxides, such as TiO_2_ and WO_3_, is difficult compared to the mentioned above oxides.

### 2.2. Sensors Based on Individual Metal Oxide 1D Structures and Their Advantages

It should be noted that sensors based on individual metal oxide 1D structures have a specific configuration and manufacturing technology, which is radically different from the traditional technology used for the manufacture of conventional gas sensors. A typical example of a sensor based on individual 1D structures is shown in Figure 4.

The interest in using individual 1D structures in gas sensors was due to the following:(*i*)It was assumed that a well-defined geometry, single-crystallinity, small diameter of 1D structures, and a large area-to-volume ratio will provide a high sensitivity of sensors [108,150]. As is known, a decrease in crystallite size and an increase in surface area is one of the basic requirements for achieving a high sensitivity of conductometric gas sensors [62,63,74,115,151]. Experiments and simulations have shown that the surface and interfacial reactions play a critical role in gas adsorption and related gas sensing processes. Therefore, the increase in an area-to-volume ratio significantly increases the number of atoms that are on the surface of metal oxides and available for reactions with gas molecules. Some of the achievements in the development of 1D structure-based gas sensors are shown in Table 1. These results indicate that 1D structure-based gas sensors can indeed be highly sensitive to various gases.(*ii*)It was assumed that due to the lack of necks and grain boundaries (see Figure 5), such sensors would solve the problem of instability and temporal drift of the parameters of polycrystalline-based thin and thick-film sensors, caused by the interaction and mass transfer on these intergrain interfaces during increased temperatures [129,149]. These processes may contribute to structural changes in the sensors [96,100,102]. The high crystallinity of the nanowires and nanobelts structure should also contribute to the improvement of thermal and therefore, temporal stability of the device parameters.(*iii*)It was assumed that response dynamics should be faster compared to their polycrystalline counterpart since there is no need for gas diffusion in the gas sensing matrix preliminary to the surface reaction [149]. The absence of nooks and crannies in nanowire-based devices contributes to the direct adsorption/desorption of gas molecules from the surface of 1D structures.(*iv*)It was assumed that through the use of individual 1D structures and self-heating effects it would be possible to significantly reduce the power consumed by the sensor, and thereby facilitate the integration of the sensors in portable systems [108,149,168,169]. It was found that the suspended nanowires (see Figure 6) are nearly ideal substances for this approach since their large length and small cross-section dramatically reduce the thermal losses to the metal leads and support.According to Meier et al. [167] and Hernandez-Ramirez et al. [158], such approach allows reducing the required power consumption from the milliwatt, in the case of micromachining technology, to the microwatt range in the case of self-heating of an individual 1D structure. It is necessary to note that the idea of self-heating was also successfully tested for thin film-based gas sensing devices (see as an example, [170]). However, the inevitable significant thermal losses to the support of the thin film sensing element impeded the implementation of this method. The major advantage of the metal oxide nanowires with respect to self-heated thin film sensors [170,171] is their small thermal capacitance and drastic reduction of thermal losses to electrodes and gas environment [168]. As seen in Figure 7, in such sensors, to achieve operating temperatures in the range of 300–400 °C, currents of only 10^−6^ A are required.In addition, self-heating effects enable an experimental methodology to improve the selectivity of metal oxide-based sensors. It was found that the thermal response time of these devices, due to a small thermal inertia, is very short—only a few milliseconds. [172]. This enables the use of higher modulation frequencies of their temperature profile, making available the direct observation of the kinetics associated with the chemical interactions between the metal oxide and the gases of interest [172]. This means that in this case, one can use the analysis of the dynamics of the sensor response, which is specific for each gas and can be considered as a marker of this gas [173].(*v*)It was assumed that the control of the shape of 1D structures (metal oxide 1D structures have a clearly defined cut depending on the synthesis conditions) would also improve the sensitivity and selectivity of the sensors. It is known that every crystallographic plane has a unique combination of structural, electronic, catalytic, and adsorption/desorption process parameters [174,175]. It is these parameters that control the operating characteristics of all types of metal oxide-based gas sensors [46,50,74,115,151,176,177,178,179,180,181,182]. Thus, the determination of crystallographic planes with optimal combinations of adsorption/desorption and catalytic parameters, and the development of methods for crystallites deposition with indicated faces can be considered as an important task for any technology used for the synthesis or deposition of metal oxides applied to a metal oxide gas sensor. Gao and Zhang [183] have summarized the recent progress in engineering crystal structures and confirmed this statement. They concluded that for the same type of metal oxides, neither surface morphology nor a specific surface area value can totally determine their sensing ability to a particular gas. Only the crystal planes exposed on the surface are a fundamental factor in determining the response of a sensor. Moreover, Gao and Zhang [183] concluded that the crystal surface with more unsaturated metal cations is the most promising approach to realize a superior gas sensing performance. According to Gao and Zhang [183], such preferred crystallographic planes are the (0001) facet for wurzite-type ZnO, (332) facet for rutile-type SnO_2_, (001) facet for anatase-type TiO_2_, (113) facet for hematite (α-Fe_2_O_3_), (111) facet for NiO, and (110) facet for Cu_2_O. Understandably, the preparation of polycrystalline metal oxides with the necessary grain faceting is difficult to control, but it is achievable for one-dimensional structures. One-dimensional structures are crystallographically perfect and have clear faceting with a fixed set of planes, which can be modified by controlling the synthesis parameters [154,184,185,186].(*vi*)It was assumed that the measurement of individual 1D structures with well-known properties can allow us to understand better the fundamentals of the gas-sensing effect [74,106,187,188]. Semiconducting one-dimensional structures of metal oxides with a well-defined geometry and ideal crystallinity can be a perfect family of model materials for systematic experimental studies and theoretical simulation of gas-sensing effects in metal oxides. The random aggregation of nanoparticles in poly(nano)crystalline MOXs, as well as the scatter in their size, make it difficult to accurately study the phenomena of gas transduction at the nanoscale. At the same time, many parameters such as film thickness, texturing, porosity, grain size, grain network, grain boundary, agglomeration, and crystallite faceting, used to characterize polycrystalline materials, lose their value for one-dimensional structures. The main structural and morphologic parameters that characterize one-dimensional structures are geometric size, characterizing the profile of one-dimensional structures, and crystallographic planes, framing these one-dimensional structures. This feature of the structure primarily refers to nanobelts. Nanobelts are thin and plain belt-type structures with a rectangular cross-section (Figure 8).At present, nanobelts are available for practically all basic oxides used in chemical sensors. There is information about the synthesis of nanobelts based on SnO_2_, In_2_O_3_, ZnO, Ga_2_O_3_, TiO_2_, etc. [78,79,80,124,189,190,191,192,193,194]. The synthesis of nanobelts can be performed using various methods [77,195,196,197], which creates good conditions for the expanding research on such nano-size materials. The minimal distance between faceting planes in nanobelts plays the same role in gas-sensing effects as the grain size in a polycrystalline material. Undoubtedly, the decreased number of parameters that control the sensor response of one-dimensional structures should contribute to a better understanding of the nature of the observed effects, since the use of 1D structures facilitates the interpretation of experimental data. Theoretical simulation of gas-sensing effects in this case can also become much simpler. [198,199,200]. As noted above, usually only one or two planes participate in gas-sensing effects in one-dimensional structures. This means that in simulations one-dimensional structures should be considered as single crystals with limited sizes.It is also important to note that currently there is already extensive experience in the synthesis of nanobelts with the growth of metal oxides in the desired crystallographic direction. For example, in Refs. [154,184,185,193] it was reported that, depending on the synthesis route, it was possible to synthesize In_2_O_3_ nanobelts with {100}, {120}, {111}, {110}, and {001} growth directions. At that point, In_2_O_3_ nanobelts grown in the {100} and {120} direction had the top and bottom surfaces being (001), while the {100} nanobelts had a side surface of (010) and a rectangular cross-section [186]. The {120} nanobelts had a parallelogram cross-section. In_2_O_3_ nano-belts grown in the {111} and {110} directions had the other set of planes; the side and top surfaces were (100) planes [127,184]. In the case of a {001} growth direction, In_2_O_3_ nanobelts were enclosed by the (100) and (010) planes [201]. The crystallographic geometry of other metal oxide one-dimensional nanostructures is presented in Table 2. Thus, the presented results indicate that when using nanobelts, we really have the ability to control the faceting planes of 1D structures of metal oxides, which is necessary both to improve the parameters of the sensors, and to better simulate gas-sensitive effects.In addition, nanobelts do not have the mechanical strength of nanotubes. Their crystallographic perfection is a very good advantage of this material. Since there are no defects in their structure, there is no problem, as is the case with nanotubes, when defects can destroy the quantum-level properties. A suitable nanobelts geometry is also an important advantage for mass production. They have a high structural homogeneity and long length. Typical nanobelts have a width of 30–300 nm, a thickness of 10–15 nm, and a length from a few micrometers to hundreds or even several thousand micrometers. [190,201]. Moreover, nanobelts are flexible and therefore can be bent 180° without damage. This fact provides additional benefits to these materials for the design of devices. The use of nanobelts, due to the specifics of their geometry, also facilitates the task of forming low-resistance contacts (see Figure 9). However, it should be noted that despite the undeniable advantages of nanobelts, nanowires are mainly used in the development of gas sensors.(*vii*)It was assumed that the use of 1D structures will allow realizing the new functionality of metal oxide conductometric sensors [108]. These expectations were partially met. In particular, it was shown that:

1D-based sensors are able to operate at room temperature [204,205]. The development of room-temperature gas sensors can have very important advantages such as a low power consumption, simple system configuration, reduced explosion hazards, and longer device lifetime;the specificity of the mechanical properties of nanowires and especially nanobelts allows the implementation of flexible sensors on their basis [206,207]. Courbat et al. [208] and Oprea et al. [209] have studied the continuous operation of a metal oxide polycrystalline gas sensor on polyimide hotplates for several months and found out that comparable or even better results may be expected from nanowire-based devices. Single-crystalline nanowires and especially nanobelts are much more resistant to the bending stress because of their flexibility;the coupling of an individual nanowire (or a mat) chemiresistor with a micro-fabricated micro-hot plate reduces the inertia of the micro-hot plate and thereby simplifies gas detection by testing the dynamics of the sensor response [210];on the base of individual nanowires, an ultra-miniature e-nose system was realized [211,212]. In its usual configuration an ultra-miniature e-nose is based on a microarray of electrodes which probe the mono-type semiconductor film, where a temperature gradient has been established. The temperature gradient induces the deviations between the responses of individual electrode pairs; and an entire array of 20 pairs of electrodes provides the required degree of orthogonality [212]. Sysoev et al. [213] showed that in such an e-nose instead of a thin film, metal oxide nanowire mats can be used. This version of the Karlsruhe Electronic Micronose (KAMINA) system is represented in Figure 10. It was established that such an e-nose had the ability to detect gases in the ppb level with an excellent stability, reproducibility, and discriminating power to effectively distinguish different target gases.

That is why since 2000, the development of gas sensors based on 1D structures has become one of the most popular areas of research in the field of gas sensors [5,86,106,144,145,149,214,215,216]. If there is an interest only in the performances of 1D-based gas sensors, then I recommend referring to the reviews [144,204,217,218,219,220,221,222,223,224,225], where such information is presented. As with conventional gas sensors, based on nanostructured crystallites, the sensing properties of individual NWs are affected by the NW diameter, the NW synthesis procedure, the feature of surface functionalization, and the reactions that occur on the NW surfaces.

### 2.3. Features of the Fabrication of Gas Sensors Based on Individual 1D Structures

As follows from previous discussions, most processes used to synthesize a one-dimensional structure are incompatible with the electrical characterization of individual nanostructures. For this purpose, an additional processing is required to remove the nanowires, nanobelts, or nanotubes from their growth substrate and then place them onto a platform that allows studying single wires and using them as a sensing element. Thus, one of the most important problems in the manufacturing of 1D structure-based sensors is their integration with electrodes [7]. An electrical contact to the sensors based on single 1D structures is often realized using a “pick and place” approach. 1D structures are first dispersed into solvents such as methanol, ethanol, isopropanol, or water in very low concentrations (Figure 11). Then, a drop of dilute suspension of 1D structures is placed on a substrate and dried [226,227]. The substrate with 1D structures are then located under an optical microscope or SEM, and the contacts are made using various methods. For example, markers are made on the substrate before applying the suspension of 1D structures, so that the position and angle of the nanowires regarding electrodes can be controlled by SEM observation. Then, a lift-off process or deposition through the mask is applied to pattern metallic contacts to the nanowires. Thereafter, 1D structures are subjected to annealing for better adherence to the electrode material. The stages of manufacturing these sensors are shown in Figure 12. This method is suitable for studying fundamental properties of the nanowires. However, it is a complicated and costly method and, therefore, not suitable for mass production of gas sensors. In some other cases, as shown in Figure 11, the metal electrodes are made before applying the nanowire to the substrate [228,229]. However, the successful placement of nanowires between the pre-fabricated electrodes requires good luck. In addition, the contact barriers between the electrodes and the nanowires are usually very high, which makes the method irreproducible and unsuitable for many device applications where low resistive contacts are required.

Undoubtedly, 1D structures can be moved to the right place using special manipulators [230] (Figure 11). For example, Li et al. [231] suggested using electrostatic forces for this. Jimenez-Diaz et al. [232] for these purposes used the AC dielectrophoretic manipulation of nanowires [228,229]. A solution, containing nanowires, is dropped on the top of the electrode array. The nanowires are aligned along the direction of an electric field. Different species such as a nanowire or contaminant species will respond to different frequencies of an AC bias. Therefore, applying an AC bias with a corrective frequency offers the advantage of selectively choosing nanowires from other contaminant species in the solution [233]. Such an approach to the formation of contacts to one-dimensional structures has several advantages, since its implementation both allows one-dimensional structures to be correctly positioned in a given place of the substrates, and can significantly reduce the manufacturing time of the sensor [232]. However, the mentioned above approaches are very laborious and require special equipment. In addition, in many cases, the transfer is possible for either too small or too large nanowires. This means that sensors based on such large nanowires will have low sensitivity. Therefore, as a rule, in order to simplify the technology of manufacturing sensors, nanowires, after application to the substrate, are preferred not to be exposed to any additional displacements.

Currently, metal strips with clearly defined shapes in the nanometer range and a high electrical quality, shown in Figure 13, are easily manufactured using various techniques such as focused ion beam (FIB) [158,159,230,235] (Figure 12), focused electron beam (FEB) [230,235], ultraviolet and shadow-mask lithography [236], and shadow mask sputtering [210,213]. Shadow mask sputtering (see Figure 12c) enables fast engineering of advanced proof-of-concept devices. However, metallic contacts directly deposited over the nanowire bundles are unstable and do not guarantee the formation of a continuous metallic layer, thus inhibiting the direct bonding of nanowires and a good electron transport. Electrical contacts, fabricated using, for example, focused ion or electron beams, are much better.

The Focused Ion Beam (FIB) is a powerful technology developed in the late 1970s and early 1980s to create a pattern, and then, to deposit materials with a resolution in the range of tens of nm. This technique is widely used in circuit editing, mask repairing, microsystem technology processes, and material characterization [237,238,239]. The basic principle of this technique is a focused ion beam of high-energy particles that scans the sample’s surface and sputters the material on the exposed area [240]. The scanning can be performed, similarly to a SEM, using electrostatic lenses and, thus, the milling occurs without the need of masks. Currently, gallium (Ga^+^) ions accelerated to 30 kV are the most used particles in the FIB technique. Gallium is a metallic element with a low melting temperature. This feature of Ga makes it possible to fabricate a durable, high-brightness, and reliable source of metal ions (LMIS) required in the FIB technique. In addition, the element gallium is located in the center of the periodic table; therefore, its momentum transfer ability is optimal for a wide range of materials. In contrast, lighter elements would be less efficient in milling heavier elements.

On the other hand, if a metalorganic compound is introduced in the beam path using the so-called microneedle-based gas delivery system, the decomposition of this compound occurs due to the interaction of the compound with both secondary electrons and ions that originated during the Ga^+^ ion bombardment (Figure 13). Part of the compound after decomposition can be deposited on the sample’s surface (ion-assisted deposition) or can reactively assist the milling process (gas assisted etching), while the other part is removed by a vacuum system. Thus, conductive and isolating materials can be easily deposited using the FIB with a nanometer accuracy [237,241].

Many types of precursor gases can be used in the FIB for the manufacture of various metal and ceramic structures. For example, the precursor gases WF_6_ (or W(CO)_6_), C_7_H_7_F_6_O_2_Au (or AuClPF_3_), (CH_3_)_3_NAlH_3_, C_9_H_16_Pt, Cr(C_6_H_6_)_2_, and TMOS+O_2_ (TMOS = tetramethyloxysilane) were used to produce W, Au, Al, Pt, Cr, and SiO_2_ layers, respectively [242]. Although the purity of the deposition is usually lower compared with conventional methods of deposition (usually the layer deposited using the metalorganic decomposition is contaminated with other elements such as oxygen or carbon from the background gas in the vacuum chamber, and elements that form the ion beam), the main advantage of this technique is its flexibility due to direct recording capabilities without using any masks [237,240].

However, it is important to note that the ion bombardment necessary to decompose the metalorganic precursor for fabricating nanocontacts can produce the damage in the nanowires [243,244]. Therefore, Hernandez-Ramirez et al. [230] believed that the development of the so-called dual or cross-beam systems (conventional FIB with a scanning electron microscope (SEM)) is very promising. This approach, enabling the in situ capture of electron images and simultaneous dissociation of metalorganic compounds with secondary electrons (SE), generated by the incident electron beam, can significantly facilitate the use of FIB nanolithography during gas sensor fabrication [188,245,246,247]. Taking into account the fact that the interaction between electrons and the sample is less destructive than using ions, the electron-assisted deposition of metals can avoid both undesired surface damage and structure modification of the nanomaterial that occur during the formation of electrical contacts on the nanostructure. Despite its advantages, the electron and ion-beam combination during deposition is sparsely used, and therefore, detailed studies of the quality of electrical contacts, formed by this method, with a purpose to avoid false interpretations of the electrical properties of nanowires, caused by the influence of contact resistance, are required [248].

It is seen that the discussed above techniques are not scalable and do not meet the requirements necessary for the transition to commercialization. A typical view of the device based on the individual nanowire is shown in Figure 14. It is seen that sensors with this configuration are really difficult to implement using methods developed for mass production. The mentioned above approaches to sensor fabrication are suitable only for research and development of prototype devices. However, device prototypes are not intended for the market. For this reason, the problem of the scalability of 1D structure-based sensors and 1D structure-based electronic circuits is one of the main ones when considering the prospects for the emergence of these devices on the market. Several approaches were developed for resolving this problem. They include self-assembly of 1D structures [249,250], roll-transfer printing [251], and microcontact/ink-jet printing [252,253]. In the self-assembly approach, metal oxide 1D structures are deposited directly by a high-voltage-driven injection nozzle onto the electrodes. During printing, the metal electrodes are precisely placed on top of the 1D structures. However, all these fabrication techniques are still in a preliminary stage of development, despite some promising results that have recently been reported [254,255,256,257,258,259,260,261].

### 2.4. Limitations of Technology Based on Individual 1D Nanostructures

#### 2.4.1. Nanowires Alignment

As mentioned earlier, at the first stage of the study of 1D nanostructures, most designers believed that the use of single crystalline individual nanowires (NWs) as a gas sensing material would significantly improve the parameters of gas and vapor sensors [129,149,150]. However, it turned out that the controlled separation, manipulation, and characterization of 1D nanostructures are not straightforward processes due to the intrinsic problems of working at the nanoscale. These procedures are complex and require well-established methodologies, which are not yet fully developed. In this regard, many attempts have been made to simplify this process. In particular, several techniques have been developed to allow the alignment or orientate 1D structure assemblies. It turned out that for these purposes one can use micro fluidics, electrostatic or magnetic fields, surface pre-pattering, self-assembly, and templating. These approaches, which allowed the design of gas sensor prototypes, were comprehensively considered in [219,249,262,263,264,265,266,267,268,269]. The advantages and disadvantages of the mentioned above methods of nanowires alignment are briefly summarized in Table 3.

As an example, in Figure 15 a diagram is shown, illustrating a method based on micro fluidics [270]. By flowing a stream of fluid across a substrate surface, nanowires can be reoriented towards the flow direction and become quasi-aligned. Parallel nanowire arrays are achieved by flowing a nanowire suspension inside the microchannel with a controlled flow rate for a set duration. The degree of the alignment can be controlled by the flow rate: The angular distribution of the nanowires narrows substantially with the increasing flow rates. The nanowire density increases systematically with the flow duration. The crossed nanowire arrays can be obtained by alternating the flow in orthogonal directions in a two-step flow assembly process. The important feature of this layer-by-layer assembly scheme is that each layer is independent of the others and, therefore, a variety of homo- and hetero-junctions, including p-n junctions, can be obtained at the crossed points. The weakness of this technique is that the area for nanowires alignment is limited by the size of the fluidic microchannels. It will be more difficult to establish a uniform shear force in a large channel.

The Langmuir-Blodgett technique is another NW assembly technology (see Figure 16). According to this technique, when a solid surface is vertically dipped into a liquid containing a Langmuir monolayer and then pulled out properly, the monolayer (Langmuir-Blodgett (LB) film) will deposit homogeneously onto the surface. Usually this technique is used for preparing thin organic monolayers. However, the LB layer of metal oxide nanowires can also be formed. To implement this method, metal oxides must meet the following requirements: (i) They must be soluble in water-immiscible solvents, and (ii) they must be able to form stable floating monolayers at the surface of the sub-phase with an internally oriented, cohesive, and compact structure that are sheer resistant. To meet these requirements, the nanowires used to form LB films are usually functionalized by the surfactants. Without functionalization, the nanowires do not form stable suspensions in the organic solvents and sink into water. A nanowire-surfactant monolayer is initially formed on a liquid (usually water) surface in an LB trough. The formed nanowire monolayers resemble a microscopic version of “logs-on-a-river”. The monolayer of the aligned nanowires is then transferred onto a substrate through vertical-dipping (i.e., LB) or horizontal-lifting (i.e., Langmuir-Schaefer (LS)) techniques. The spacing between the parallel nanowires can be adjusted by the lifting speed and by the pressure of the monolayer compression.

The features of the roll printing technique and contact printing technology are shown in Figure 17 and Figure 18, respectively.

In regards to the orientation of nanowires in an external electric field, this effect is caused by the fact that all metal and semiconductor nanostructures have a dielectric constant that differs from that of the surrounding medium (air or liquid). The resulting image forces produce a rotational moment, causing the nanostructure to align in the direction of the electric field. Ruda and Shik [272] showed that the alignment depends dramatically on the relationship between the nanowire length *L*, separation between electrodes *a*, and electrode width *b*, as well as on the nanowire dielectric constant ε compared to that of environment ε_0_.

However, it must be recognized that all fabrication routes, developed for nanowires alignment, are not perfect processes and do not apply to fully completed technologies. Moreover, the listed techniques work with an array of 1D nanostructures, and they do not guarantee the fixation of individual 1D structures in the required place. Thus, the use of 1D nanostructures in real devices is still at a preliminary stage and needs a breakthrough in order to integrate them with low-cost industrial processes [149,267,269,273].

#### 2.4.2. Sensor Performances

The low sensitivity of most individual 1D-based sensors in comparison with conventional nanocrystallite-based sensors can also be attributed to the shortcomings of these devices. As in polycrystalline sensors, the sensitivity of 1D-based sensors depends on the diameter of nanowires or nanobelts (see Figure 16); the smaller the diameter, the higher the sensitivity [128,158,159,274]. The experiment and simulation have shown that in order to achieve a maximum response, the NW radius must be compared with the Debye length (*L*_D_). For example, Hernandez-Ramirez et al. [158,159] found that in real 1D SnO_2_ structures the Debye length is ~10–15 nm and, therefore, the performance of nanowire sensors can be significantly improved only if the diameter of the nanowires is less than 25 nm. However, as a rule, nanowires used for gas sensor fabrication have a diameter in the range of 50–900 nm, while the grain size in polycrystalline-based sensors does not exceed 10–30 nm. Research carried out by Li et al. [126,154] and Zhang et al. [155] confirmed this conclusion. They established that an extremely low detection limit was achieved for In_2_O_3_ and ZnO nanowires with a diameter of only 10 nm. Unfortunately, currently nobody wants to work with individual nanowires with a size less than 30–50 nm [128,158,159,275,276]. Firstly, the controlled growth of very thin nanowires and the fabrication of devices based on such nanostructures is still an experimental problem [108,277]. Secondly, separation and manipulation of such small objects is too difficult [158,159]. In addition, there are technological difficulties with the growth of such thin nanowires (*d* < 10 nm) with a longer length. However, if the nanowire is short, the length of nanowires can be insufficient to create a bridge between two bonding pads on the measuring platform [276].

Kolmakov et al. [278] have found that one of the most promising solutions to this problem is the fabrication of a single-crystal quasi-1D chemiresistor with one (or a few) very thin segments, which adheres to the *r ~ L_D_* condition, where *r* is the radius of the nanowire in this place. Since the narrow segment(s) will control the electron transport and sensing performance, such a device would have all the advantages of ultra-thin single crystal nanowires (see Figure 19). Studies carried out by Dmitriev et al. [274] showed that the narrow segments serve as ideal “necks,” as observed between particles in conventional polycrystalline thin film gas sensors, but provide the significant advantages of greater morphological integrity and stability. Reports of the methodology of the segmented oxide nanowire (SN) controllable growth via a programmable change in the local vapor supersaturation ratio (Sn*_x_*O*_y_* (*x*, *y* = 1, 2 …)) in the vicinity of the wires during their vapor solid growth have been published in [278]. This is an interesting but complicated approach to the manufacture of 1D-based sensors.

According to Kolmakov [108], the segmentation of NWs can also be organized using approaches such as:increased amount of defects and, therefore, increased scattering of the carriers in a selected area;lateral inhomogeneity in doping and, therefore, formation of Schottky-like junctions along the length of the nanowire; andnarrowing of the conductive channel of the nanowire as a result of sputtering.

The creation of a reliable low-resistance electric contact with such thin nanowires is also an essential problem [273,279], since the quality of the contacts directly influences the performance of the sensors. The reduced contact area between metal electrodes and NWs increases the contribution of the electrical properties of the contact in the results of nanowire characterization, which may hide the phenomena occurring on the NW surface due to the interaction with a gas environment [158,230,244,280,281]. Therefore, Ebbesen et al. [243] and Hernandez-Ramirez et al. [230] believed that in order to reduce the influence of contacts on sensor readings, it is necessary to use a four-probe contacts configuration of the sensors (see Figure 20a). It is also necessary to monitor the dissipated power on the contacts in the process of testing. Even with powers exceeding 1 μW of heat dissipation, local heating is possible, which can originate many physical problems related to material diffusion, transformation, and fusion, that must be avoided to ensure nanosensor stability. Figure 20b shows the final electrical breakdown consequence of this phenomenon. Hernandez-Ramirez et al. [158] have also noted that before arrival to this failure state, the electrical characteristics drifted and showed a strong degradation and non-repeatability. Comini et al. [5] believed that the exact technical difficulty, connected with the fabrication of reliable low-resistance electrical contacts on individual 1D nanostructures, is one of the reasons, limiting the number of works on this topic.

#### 2.4.3. Stability and Reproducibility

It should be noted that despite the seemingly indisputable advantages of 1D structures for the development of gas sensors with increased stability, the problem of thermal instability of 1D structures cannot be neglected. If we analyze the results confirming the increased stability of sensors based on 1D structures compared to conventional metal oxide gas sensors (see for example, Figure 19), we will see that all statements, regarding an experimental confirmation of the better stability of the 1D structure-based gas sensors, relate to sensors fabricated using nanowires with the diameter significantly exceeding crystallite sizes in polycrystalline metal oxides. For example, the results presented in Figure 21 were obtained for nanowires with a diameter of 200–900 nm and nanocrystallites with a diameter of ~4 nm.

Unfortunately, at present, as we mentioned above, nobody wants to work with individual 1D structures with a characteristic size of less than 20–30 nm [128,158,159,275,276]. Of course, in the future, attempts will be made to increase the response of 1D-based gas sensors by reducing the diameter of the nanowire. In this case, we must be prepared for the fact that if we have the opportunity to fabricate sensors based on nanowires with a diameter of less than 10–15 nm, it is quite possible that we will also face the problem of reducing the stability of these devices. For example, studies carried out on the basis of metal and semiconductor nanowires, showed that 1D structures with a characteristic size in the range of 10–20 nm are also unstable [109,282,283,284,285,286]. With a decrease in the diameter of nanowires their melting temperature drops sharply, just as in the case of nanocrystallites [287,288]. For example, the melting of a Ge nanowire with a diameter of 55 nm begins at a temperature of about 650 °C, while the melting point of bulk Ge is 930 °C [282]. Given the tendency of thin nanowires to be cut due to melting at a relatively low temperature, we must recognize that sensors, in which such a phenomenon is possible, cannot be considered devices with increased stability. This limitation, undoubtedly, should be taken into account during the analysis of the prospects of 1D structures using conductometric gas sensors developed for operation at elevated temperatures. This limitation may be especially relevant in the development of self-heating sensors [102], where even small variations in the diameter of nanowires and contact resistance or the uncontrolled increase in the current can cause local heating to temperatures close to or above the melting temperature (*T*_m_). One should also to take into account that due to the nonlinearity of the I(V) characteristics and because of the reducing resistance of nanowires during the interaction with reducing gases, the Joule power dissipated by the nanowire can be increased even under a constant bias voltage [289]. These processes form a positive feedback and, therefore, they can ultimately destroy the nanowire resistor. This effect is usually observed for metal and highly doped semiconductor nanowires with moderate melting points [290,291].

The instability associated with surface diffusion of the electrode material along the nanowire and the migration of surface clusters used for surface functionality can also be significantly increased; 1D structures do not contain edges and stages, which can serve as particle pinning centers that hinder the diffusion of noble metal clusters on the surface of metal oxide supports.

The reproducibility of the parameters of the 1D nanostructure-based gas sensors should also be improved [273]. Due to the sample to sample variation (see Figure 22), it is very difficult to produce individual 1D structure-based gas sensors with identical parameters. A small change in the size and shape of 1D structures leads to a sharp change in electrophysical, electronic, and surface properties, which will undoubtedly be accompanied by a significant change in sensor performances. However, this is unacceptable for the sensor market. All sensors must have the same parameters.

The lack of technology for the synthesis of well-controlled one-dimensional structures with an industrial-scale production rate is also a significant limitation [130]. However, the hydrothermal method can be considered as a potential scale-up method for the synthesis of 1D nanomaterials.

There are also difficulties in growing the 1D structures, faceted by crystallographic planes with more unsaturated metal cations. As previously shown, these planes can provide maximum sensitivity for gas sensors. Gao and Zhang [183] noted that the controlled synthesis of metal oxide nanomaterials with exposed high-energy facets is a difficult task since facets with a high surface energy usually grow rapidly and eventually vanish. Although there have been several successful cases, there is still a lack of understanding of the growth mechanism of high-energy facets. More in-depth research should be carried out to investigate the mechanism and find out viable synthesis methods. There are also questions regarding the long-term stability of high-energy facets during the gas sensing process. High-energy crystal surfaces are extremely active. Therefore, it is unclear how long a gas-sensitive metal oxide-based 1D nanostructure with exposed high-energy facets can remain active and exhibit excellent performance?

### 2.5. How to Improve 1D Structure-Based Gas Sensors

#### 2.5.1. Sensors Based on 1D Structures Array

The study showed that the application of an array of 1D nanostructures, placed in the form of monolayer mats (see Figure 23a), allows the use of a standard thick-film technology for gas sensor fabrication. It also gives the opportunity to eliminate many of the technological difficulties associated with the manufacture of 1D nanostructure-based sensors and makes it possible to already realize now a number of indisputable advantages of 1D nanostructures usage in gas sensors. That is why this approach is currently the most common in the manufacture of gas sensors based on 1D nanostructures [213]. However, in this case, such gas sensors do not have a fundamental difference from traditional polycrystalline sensors manufactured by the thick-film technology. Impedance spectroscopy studies showed that the gas-sensing mechanism for sensors on the base of networked 1D nanowires involves changes in both the nanowire resistance, and the resistance of inter-nanowire interface [293]. This situation is shown in Figure 23b. The only advantage of such sensors is a large porosity, that is, the better gas permeability of the sensing layer and, as a result, faster response.

Nanowires can be converted to a gas sensitive layer using several methods. The 1D metal oxide NWs mat can be converted to a gas sensitive layer using several methods. Deposition of a slurry containing NWs onto the electrodes and subsequent heat treatment is the simplest method to fabricate the gas sensor [295,296,297]. However, in this case, the use of transfer media, in most cases liquid, poses potential harm to the unique properties of the nanowires, such as surface reactivity. As is known, the surface cleanliness of a nanowire is crucial for gas sensor performances. This means that the reduction or elimination of post-processing of the nanowires during gas sensor fabrication is the best solution of this problem. In this regard, the integration of nanowires into devices by the direct growth of nanowires in the required place of the substrate has an obvious advantage over transfer methods. In this case, the mat-type films can be prepared by simply pressing long NWs grown on a substrate. However, both these methods do not provide a reproducible deposition of the NW network onto a defined area, especially in cases where the gas-sensitive layer must be formed on a sub-millimeter-scale or micrometer-scale area [296]. In addition, the sensor stability against vibrational environments can be deteriorated due to the poor adhesion between the NWs and the electrodes.

The direct growth of oxide NWs on metal or metal oxide electrodes can be an alternative method for manufacturing well-defined metal oxide NWs-based gas sensors [153,294,298]. A strong adhesion between the NWs and electrodes significantly improves not only the electrical contacts between the NWs and electrodes, but also the operation stability against mechanical vibration. In addition, highly miniaturized sensors can be fabricated by the selective growth of NWs within a defined area. Below we will consider these methods in more detail. However, to begin with, we note that the functional arrays of various metal oxide nanowires could be synthesized using either the template-free method or template-assisted strategy. The template-free synthesis provides a fast enough growth of 1D nanostructure arrays of a big group of metal oxides, suitable for chemical sensors, such as ZnO, Ga_2_O_3_, TiO_2_, and In_2_O_3_. Their growth is controlled by adjusting process parameters such as the temperature, process duration, and the type and the concentration of the precursor. In contrast, the template-assisted method, due to the possibility to control the template geometry and pattern distribution, could be more effective in adjusting the morphology and size of nanostructures (length, density, and diameter). The patterned templates as a unique mask could also endow the ability to define the shape and the position of nanoarrays. However, it must be borne in mind that the template-assisted method has limitations on growing nanowires with a minimum diameter. The template-free method is free from this drawback. However, when using the template-free method, due to not using the uniform growth, poor control over the density and distribution of NWs is observed; 1D structures can have a wide variation in their geometric parameters such as length, thickness, or diameter. This significantly limits the possibility of manufacturing sensors with reproducible parameters.

#### 2.5.2. Sensors with Vertically Oriented Nanomaterials

One of the progressive approaches to the fabrication of gas sensors, based on 1D structures array, involves growing parallel NWs or nanotubes on the substrates in the perpendicular direction to the surface (see Figure 24). In this case, although the 1D structures array is used, this configuration makes it possible to realize the advantages of the sensors characteristic of devices based on individual 1D nanostructures. In [106,299,300], the application of this approach to the development of devices from TiO_2_ nanotubes was described. This technique is similar to the manufacture of thin films, but promises higher accuracy in making the columnar nano-elements. Contacts can also be made using electrochemical deposition methods [279].

A description of other approaches to the formation of vertically oriented nanomaterials as applied to ZnO can be found in [219,301,302,303]. For example, such structures can be created using template-assisted techniques, such as filling the membrane pores of porous silicon or anodized alumina [304,305]. This template-based method for nanowire array fabrication provides a low cost technology acceptable for large-area processes. This approach is one of the feasible techniques to form well-defined lateral or vertically oriented nanowires arrays, since it is easy to form electrodes to selected nanowires. However, as a rule, nanowires, grown using the template-based method, are polycrystalline and, in principle, do not apply to the nanomaterials considered in this article. As we indicated earlier, in this article we consider single-crystal 1D nanostructures.

The view of vertically oriented NWs really related to 1D nanomaterials is shown in Figure 25c. The method, developed by Wei et al. [302] for growing vertically oriented ZnO NWs, combines a laser interference pattering technique and hydrothermal synthesis. A hole array is fabricated on the surface of the substrate through laser pattering. The NWs are then formed from these holes in a mixed growth solution, containing Zn(NO_3_)_2_ and hexamethylenetetramine (HMTA, (CH_2_)_6_N_4_) by using a hydrothermal method (Figure 25a,b). However, in order to look this way, NWs must have a diameter much larger than 500 nm, which does not contribute to the high sensitivity of gas sensors made on their basis. With a smaller diameter, vertically oriented nanowires do not look so perfect. In particular, Figure 26 demonstrates vertically aligned ZnO nanowires directly synthesized on fluorine-doped tin oxide-coated substrates using the Chemical Vapor Deposition (CVD) method. Moreover, even in this case, the diameter of nanowires was quite large. The top and bottom diameters of the needle-shaped ZnO nanowires were around 100 nm and 1 µm, respectively [306].

However, in reality, based on 1D nanowires vertically grown without any template, it is difficult to implement the structures shown in Figure 24. The main difficulty lies in the formation of contacts to the top of vertically grown nanowires. The deposition of a metal film by physical vapor deposition, will not work as the metal seeps through the voids between the NWs and shorting with the bottom electrode. Several approaches have been proposed to form such contacts. For example, to eliminate the shorting with the bottom electrode, the NWs can first be embedded in a dielectric layer such as a polymer (see Figure 27). Then, planarization is performed to obtain NWs of the desired length. After etching the polymer with purpose to expose the tips of the NWs, a metal film is deposited on top as a contact pad. The experiment has shown that oxygen plasma etching is the best method for polymer etching and NWs cleaning. After metal deposition, the polymer is completely removed and frees the NWs to interact with the atmosphere [307,308]. As we can see, this is a rather long and laborious process.

In order not to use any treatments that affect the surface properties of metal oxides, Parthangal et al. [309] proposed a different approach. An electrical contact with the top of the nanowire array is fabricated by creating a continuous conductive film through the electrostatic attachment of conductive gold nanoparticles exclusively onto the tips of nanowires (Figure 28). Au nanoparticles were generated through an aerosol spray pyrolysis method. The droplet-containing flow was passed through dehumidifiers and then into a tube furnace maintained at 600 °C to thermally crack the precursor and form Au particles. The particles were then positively charged with a home-made unipolar charger and introduced into an electrostatic precipitator containing the substrate with the grown nanowire arrays of ZnO. A high negative electric field of −10 kV/cm was applied to drive the particle deposition. Parthangal et al. [309] stated that this assembly approach is suitable for any nanowire array that requires an upper contact electrode. However, we must admit that this method of forming contacts is also not simple and does not provide a good process performance (see Figure 28).

Radha et al. [310] developed another interesting approach to forming contacts to vertically grown nanowires that does not require additional operations. It is schematically represented in Figure 29. To implement it, they used a chemically synthesized single-crystalline Au microplate as the top electrode. The contact is electrically activated, and the formation of the contact is mainly due to electromigration. With this approach, the electrode could ohmically contact several thousand nanowires at once. This method has been developed for InAs nanowires. However, it can undoubtedly be used as applied to metal oxide nanowires, though only for research purposes.

The approach proposed by Choi et al. [134] and Ahn et al. [311,312] was also of interest for the gas and vapor sensor design (see Figure 24). Figure 30b,d shows side- and top-view scanning electron microscope (SEM) images of ZnO nanowires grown on patterned electrodes. ZnO nanowires, grown only on the patterned electrodes, have many nanowire/nanowire junctions as seen in Figure 30c. These junctions act as an electrical conducting path for electrons. The device structure is very simple and efficient for gas sensitive elements forming, because the electrical contacts to nanowires are self-assembled during the synthesis of nanowires. In addition, growing nanomaterials directly onto the sensor chips simplifies the manufacture of sensors. Ahn et al. [311,312] asserted that this method of on-chip fabrication of nanowire-based gas sensors is scalable and reproducible. Furthermore, it integrates stable nanomaterials onto the electrode surface without using any binders for fixing nanomaterials onto the electrodes. A more detailed description of the sensors based on vertically oriented nanomaterials can be found in [220]. It is important to note that this method of manufacturing sensors does not require any manipulation of nanomaterials, and therefore, in their manufacture, one can use the technological modes that allow growing nanowires of a small diameter. This means that sensors manufactured in this way can have very high sensitivity, which was experimentally confirmed [130,219,220,313]. For example, Figure 31 shows the results reported by Yuon et al. [313]. It is seen that sensors really show very high sensitivity. High sensitivity appears to be due to the small area of inter-nanowires contacts. With a sufficiently large diameter of NWs, only the presence of the adsorbate-modulated Schottky barriers [108], formed between individual nanowires, can explain high sensitivity.

However, despite the general progress in on-chip fabrication of metal oxide-based nanomaterials, few of them can be directly grown on substrates/electrodes. In this regard, ZnO is the best material for these purposes. It should also be taken into account that NWs in such structures can form inter-nanowires contacts without sintering. This means that with any vibration of the sensor, the area and the number of such contacts can change, leading to a change in the basic sensor characteristics. According to McAleer et al. [314], a significant reduction in the number of contacts in the NWs networks, involved in the percolation-dependent transport, compared with polycrystalline metal oxides, inevitably should worsen also the stability (thermal, mechanical, and chemical), as well as the overall performance of the devices, fabricated using the indicated approach [108].

#### 2.5.3. Sensors with Horizontally Oriented Nanomaterials

The experiment showed that from the point of view of the stability of sensors parameters, a more reliable configuration is the structure shown in Figure 32, when the nanowires grow and form a bridge between the electrodes, fabricated on the substrate [163,219,315]. However, to implement this configuration, NWs should be sufficiently long to create a connection between the two electrodes. Therefore, the distance between the electrodes should be extremely small. This will facilitate both the formation of the bridge and will allow the formation of such bridges using nanowires of a smaller diameter, which is necessary to achieve high sensitivity. Another important step for the NWs growth is the deposition of a catalytic film on the sidewall of electrodes. This step allows the NWs to grow first on the catalytic film and then pass through the trench to the opposite side of the electrode. Unfortunately, it is difficult to control this process, which introduces an element of randomness and uncertainty in the properties of sensors manufactured using this technology.

Additionally, noteworthy is the approach proposed by Kim and Son [316]. They adapted a step edge decoration (SED) method [317], as a bottom-up technique to fabricate a lateral nanowire array (see Figure 33). Using this technique, high quality ZnO nanowires with a diameter of about 20 nm and a regular interval of about 80 nm were fabricated on a sapphire substrate. Gas sensors based on these structures demonstrated a high response to ethanol (S = R_g_/R_a_ ~170 for 200 ppm). Despite the possibility of creating gas sensors based on individual nanowires without using any operations related to their separation and manipulation, this technology is still too expensive for large-scale use.

In this regard, the methods described in [318,319,320] are more suitable for forming a lateral nanowire array. These methods are based on creating conditions for the growth of nanostructures along the surface of the substrate. For example, the crystallographic property of the substrate should be suitable for the orientation growth of growing NWs along the substrate surface. Using this approach, Nikoobakht et al. [318,319] successfully fabricated horizontally aligned ZnO NWs on an α-plane sapphire (1$\bar{12}$0). ZnO NWs are grown along the [1100] direction, in which the lattice parameters of c-plane ZnO NWs and α-plane sapphire are similar. In this architecture, NWs can be grown on the places where the nanodevices could be subsequently manufactured. In this case there is no need to transfer NWs to a different surface or align them. With the use of only three photolithographic steps, this technique allows the industrial-scale production of nanodevices (see Figure 34). First, an α-plane sapphire surface is patterned with gold nanodroplets. Next, small-diameter zinc oxide NWs are grown selectively on the predefined gold sites. A growth direction of the NWs is controlled using the anisotropic crystal match between zinc oxide and the underlying substrate. Unfortunately, it is very rarely possible to fulfill the condition for horizontal growth, because substrates should exhibit a good lattice mismatch with the growing NWs. This significantly limits the applicability of this method. In addition, the reality is not so good. The method suggested by Qin et al. [321] allows circumventing this limitation. Qin et al. [321] showed that by applying the combined effect of a ZnO seed layer and a catalytically inactive layer (Cr or Sn), one can fabricate horizontally aligned ZnO NWs on various organic and inorganic substrates with single crystal, polycrystalline, and amorphous structures. Wang et al. [322], when this method was employed, fabricated the aligned SnO_2_ NWs and showed that gas sensors based on these structures can have a high sensitivity to NO_2_.

#### 2.5.4. Large-Scale Fabrication of Nanowire Networks-Based Gas Sensors

As we can see, quite a lot of different methods of direct growth of NWs have been developed to date, allowing the production of 1D nanostructure-based gas sensors. However, to date, a reliable technological route for the production of large-scale gas sensors based on the NW networks for commercial purposes has not yet been developed. Hwang et al. [153] suggested a facile route to fabricate single crystalline SnO_2_ NW networks-based gas sensors on a large scale. The SnO_2_ NW networks were grown on 16 different sensor elements via a single step vapor phase reaction using laser-scribed Al_2_O_3_ substrates with patterned Au catalyst layers, Au electrodes, and Pt heaters (Figure 35). NWs were several tens of micrometers long and 30–100 nm thick. Hwang et al. [153] showed that fabricated sensors had high sensitivity and selectivity to NO_2_ and C_2_H_5_OH (see Figure 36). At the same time, they did not show how identical the parameters of all these 16 sensors were. Without this, one cannot talk about a large-scale production. It should be noted that the nanowire networks shown in Figure 35c looks like a loose structure in which the position of the nanowires relative to each other is not fixed. In this situation, doubts arise about the stability of this structure and its electrophysical and gas sensing properties when exposed to an external vibration.

## 3. 2D Nanomaterials

### 3.1. Introduction

The use of 2D nanomaterials is another promising area in the development of gas sensors. 2D nanomaterials are objects that in one direction have a size in the range of up to 100 nm. The most famous 2D nanomaterials are graphene [8,323], black phosphorus or phosphorene [324], silicene, borophene, boron nitride [325], and transition metal dichalcogenides (TMDs) [326], which have a layered structure and allow the formation of atomically thin 2D nanomaterials of a large area. 2D materials are characterized by a high degree of anisotropy and chemical functionality. All these nanomaterials differ in their mechanical, chemical, electrophysical, surface properties, as well as in size, shape, biocompatibility, and stability to the influence of external factors. However, their low-dimension nanostructure gives them some common characteristics. 2D nanomaterials are the thinnest materials known, which means that they have the highest specific surface areas among all known materials. As a result, they can adsorb large numbers of gas and drug molecules and provide excellent control over the kinetics of surface reactions. In addition, the thinness of these nanomaterials allows them to respond rapidly to external signals such as light or chemical and biological agents. This characteristic makes these materials invaluable for applications requiring high levels of surface interactions on a small scale. An experiment showed that these properties make 2D nanomaterials suitable for a wide range of applications, such as drug delivery, energy conversion and storage, electronics, optoelectronics, biosensing, various biomedical applications, and design of chemical sensors, including gas sensors, etc. [327,328,329,330,331,332,333].

### 3.2. Advantages of 2D Nanomaterials for Gas Sensor Design

The main advantages of 2D nanomaterials for use in gas sensors include:
2D layered nanomaterials, as well as 1D nanomaterials, possess a large surface area and high surface-to-volume ratio (see Figure 37), allowing more atoms to interact with the atmosphere. This is especially important for gas sensors, since such a structure of a gas sensitive material facilitates surface reactions, such as adsorption, chemisorption, and heterogeneous catalysis, controlling gas sensitive effects [324,334].Unique structures of 2D nanomaterials can provide these materials with properties that cannot be achieved with conventional bulk structures.The 2D layered structure has an increased mechanical stability, which is very important when developing sensors on flexible substrates [336,337].Bandgap engineering is another advantage of 2D nanomaterials for gas sensor applications. It is assumed that bandgap engineering via a thickness control at the atomic level will make it possible to control the electronic and adsorption/desorption properties of gas sensing materials [324].Due to their relatively larger lateral size, 2D nanostructured materials offer a better conformal contact with the electrodes in comparison with 1D nanomaterials.The possibility of assembly into three-dimensional (3D) architectures can also be attributed to the advantages of 2D nanomaterials [338].In addition, it should be borne in mind that in 2D nanomaterials only one crystallographic plane is involved in gas-sensitive effects. This, as well as in the case of 1D nanostructures, significantly simplifies the modeling of processes, occurring on this surface, and contributes to a better understanding of the nature of gas-sensitive effects [339]. Moreover, when the crystallographic structure of 2D nanomaterials is correctly selected, then this factor can contribute to improved sensor selectivity.Different composite materials based on two-dimensional materials is another opportunity for creating gas-sensitive materials with unique properties that are not accessible in a 2D material or 3D structures based on one oxide material [340,341]. The experiment showed that, in addition to metals, polymers, and metal oxides, other 2D nanomaterials can be used to form 2D-based composites. Layered combinations of different 2D materials are generally called van der Waals heterostructures or hybrid two-dimensional materials. Hybrid two-dimensional materials are considered to be advanced multifunctional materials that have outstanding physical and chemical properties important for various applications [342,343]. In particular, Shanmugasundaram et al. [344] and Yang et al. [345] believed that gas sensors based on such heterostructures can overcome the disadvantages inherent in gas sensors based on simple materials such as low selectivity. Some approaches that can be used in the formation of 2D-based heterostructures are described in [346].

Currently, based on these 2D nanomaterials, a large number of prototypes of gas sensors have been made, which have increased sensitivity to various gases. An analysis of these results can be found in numerous reviews devoted to the consideration of these materials [323,326,334,345,347,348,349,350,351,352,353,354].

### 3.3. Features of 2D Nanomaterials Synthesis

Studies have shown that in addition to traditional layered 2D nanomaterials such as graphene, black phosphorus or phosphorene, silicene, borophene, boron nitride, and transition metal dichalcogenides (TMDs), metal oxides may also have 2D structures [351,355]. It was established that some metal oxides such as Ga_2_O_3_ and metal trioxides with a general formula of MO_3_ (M = Mo, Ta, W, etc.) have a layered structure in hydrate or anhydrate phases [356]. For example, MoO_3_ has a layered structure in which each layer is predominantly composed of a distorted MoO_6_ octahedra in an orthorhombic crystal (space group P_cmn_) [357]. The α-V_2_O_5_, the most stable phase in the vanadium oxide family due to its highest oxidation state of vanadium, also has a layered structure and crystallizes with an orthorhombic unit cell structure (space group: P_mmn_). Each layer is composed of a distorted trigonal bipyramidal polyhedral where O atoms locate around the V atoms [358]. To form 2D nanomaterials based on such layered metal oxides, one can use methods commonly used in the formation of graphene, phosphorene, and transition metal dichalcogenides (TMDs) [324,348,359]. Thus, Ji et al. [360] combined grinding and sonication to exfoliate bulk α-MoO_3_ crystals into single- and few-layer nanosheets. A surfactant was used to stabilize them. It is important that nanosheets of the aforementioned oxides with oxygen terminated basal surfaces are stable both in air and in water.

Perovskites are another large family of layered 2D metal oxides [361]. Layered perovskites consist of ABO_3_ layers interleaved by thin sheets of motifs with A and B cations of possibly different sizes. Three main layered perovskites are the Aurivillius (AU), Dion-Jacobson (DJ), and Ruddlesden-Popper (RP) phases [362]. The general formula of the AU phases is [Bi_2_O_2_]-[A_(n−1)_B_n_O_3n+1_]. Bi_2_WO_6_ and SrBi_2_Ta_2_O_9_ are examples of such oxides. The general formula of the DJ phases is MA_(n−1)_B_n_O_(3n+1)_. Metal oxides LaNb_2_O_7_, (Ca,Sr)_2_Nb_3_O_10_, and La_2_Ti_2_NbO_10_ have such a formula. The RP phases include compounds such as SrLaTi_2_TaO_10_ and Ca_2_Ta_2_TiO_10_. They can be directly synthesized into a 2D form using layer-by-layer deposition techniques with possible stabilization steps. If the perovskites bulk crystals are stratified, similar to graphene or TMDs, 2D perovskites can be directly exfoliated from their naturally stratified phases. However, indicated perovskites were not used in the development of gas sensors. Many of the parent layered perovskites are highly polarizable and form the basis of dielectric and ferroelectric materials [362].

As for the traditional metal oxides used in the development of gas sensors, such as SnO_2_, ZnO, In_2_O_3_, etc., they do not have a natural layered crystal phase. Therefore, for the formation of 2D nanostructures of these metal oxides, one has to use other approaches based on the development of special synthesis modes. Using this approach, 2D nanomaterials based on oxides such as ZnO [200,363,364,365], SnO_2_ [366,367], WO_3_ [368], CuO [369], Co_3_O_4_ [370,371], In_2_O_3_ [372], and V_2_O_5_ [373] were formed. It was established that for these purposes it can be used as either the hydrothermal or solvothermal synthesis [200,363,364,365,366,367,368], as well as precipitation [374], layer-by layer deposition, and sonochemical processes [375]. The appearance of such structures is shown in Figure 1c,e. Typically, these structures are quasi-single crystalline structures.

### 3.4. Gas Sensors Based on 2D Nanomaterials and Their Limitations

#### 3.4.1. Sensors Performances

The parameters of gas sensors developed on the basis of 2D metal oxides are shown in Table 4. As one can see, sensors based on 2D nanomaterials have the characteristics typical of metal oxide-based gas sensors. The maximum sensor response, depending on the material used, its structural parameters, and the nature of the detected gas, is observed in the temperature range from RT to 400 °C. The best samples are also fast-acting and exhibit increased sensitivity.

It should be noted that both in the formation of sensitive layers, and in the fabrication of sensors based on 2D nanomaterials, in order to optimize their parameters, all approaches, developed previously for conventional metal oxides and other 2D nanomaterials, can be used [376]. They are bulk doping or surface modification with noble metals [377] and metal oxides [378,379,380]. As with conventional metal oxides [115,123], these approaches provide a significant improvement in sensor parameters such as sensitivity, selectivity, and the time of sensor response and recovery (see Figure 38). For example, Yin et al. [379] showed that the response of the WO_3_ nanosheet-based sensors coated with SnO_2_ nanoparticles to a 50 ppm acetone vapor was 10 times higher than that of the pristine WO_3_ nanosheet gas sensor. However, it must be borne in mind that despite the progress made, the efficient integration of 2D functional layers with a three-dimensional sensor platform remains a significant challenge, limiting device performance and circuit design [381].

Bag and Lee [389] showed that the parameters of 2D structure-based gas sensors can also be improved through the formation of various types of heterostructures, such as 2D-0D, 2D-1D, 2D-2D, and 2D-3D heterostructures. However, they also recognize that despite their many advantages, the development and manufacture of 2D nanostructured material-based gas sensors using heterostructures still face problems. Too many technological difficulties arise in the preparation of gas-sensitive materials based on such heterostructures. At that point, the formation of 2D-2D heterostructures is of particular difficulty, since in this case it is necessary to layer one 2D nanomaterial onto another with the formation of a close contact between them. A description of possible approaches to the formation of such structures can be found in [346].

At the beginning of this section, we noted that one crystallographic plane is involved in gas sensitive effects in 2D nanomaterials, which makes it possible to determine the crystallographic planes most suitable for constructing gas sensors with improved parameters. Unfortunately, due to the difficulties in forming nanosheets with the required crystallographic orientation, research in this direction is almost not carried out. Only Xu et al. [200] in their work have compared the gas-sensitive characteristics of ZnO nanosheets (NSs) with (0001) и (1$\bar{01}$0) planes and established that the ZnO NS (0001) sensor shows a better performance than the NS (1$\bar{01}$0)-based sensors (see Figure 39). This means that the 2D ZnO structures faceted by the (0001) plane are preferred when developing gas sensors.

#### 3.4.2. Technologies of Sensor Fabrications and Their Limitations

Despite the progress made in the development of sensors based on 2D metal oxides, it must be recognized that these sensors are still inferior in parameters to many conventional metal oxide nanocrystallite-based gas sensors. Some studies [360,366] point to the significant superiority of 2D nanomaterial-based sensors over conventional polycrystalline gas sensors. However, as a rule, in these articles, the conventional polycrystalline gas sensors that are used for comparison do not have the optimal structure of the gas-sensitive layer. For example, Lou et al. [366] compared sensors based on 2D SnO_2_ nanosheets with a thickness of ~15 nm with sensors, fabricated using agglomerated SnO_2_ powders with a crystallite size of more than 20 nm.

From our point of view, the advantages of conventional gas sensors based on polycrystalline material in comparison with 2D nanomaterial-based devices are due to the following reasons:

Firstly, there are technological difficulties in the fabrication of gas sensors based on 2D nanomaterials [360,390,391,392]. For example, if 2D nanomaterials are in the form of a large nanosheet, then in the case of a thick-film technology used for forming a gas-sensitive layer, there is a threat of the formation of the layer, consisting of strongly agglomerated 2D nanosheets. This makes it difficult for the gas to penetrate the inter-nanosheet space. Additionally, this further reduces the sensitivity and significantly increases the response and recovery times. Wherein, unlike the conventional material, i.e., polycrystalline metal oxides, we cannot use the mechanical milling of agglomerated materials, since milling will be accompanied by crushing of 2D nanomaterials.

Secondly, it is very difficult to grow 2D nanomaterials, which are large but small in thickness, using traditional approaches to the synthesis of metal oxides. When nanosheets become large in size, then their thickness becomes more than 10–50 nm, which does not contribute to achieving high sensitivity of the sensors. If during the synthesis of 2D nanomaterials we try to limit the thickness of the plates, then in the end we get a material that is slightly different from ordinary nanocrystallites used in conventional gas sensors.

Some problems associated with the excess thickness of nanosheets have been resolved on the basis of new technological approaches [348,393]. It was shown that with an appropriate optimization of the process, thin nanosheets can be synthesized using a wide variety of techniques such as hydro/solvothermal synthesis [335], the interface-mediated synthesis method [394], 2D-oriented attachment method [395], 2D-templated synthesis [396], self-assembly of nanocrystals [395,397], and the on-surface synthesis method [398]. Using this strategy, large 2D structures of TiO_2_, ZnO, Co_3_O_4_, WO_3_, Fe_3_O_4_, V_2_O_5_·0.76H_2_O, and MnO_2_ with a thickness ranging from 1.6 to 5.2 nm (corresponding to 2−7 stacking monolayers) have been synthesized [355,373]. It is important to note that metal oxides with such a thickness should have unique properties in terms of gas sensitive effects. For example, five-atomic-layer-thick SnO_2_ sheets (0.66 nm) have a 40% surface occupancy by highly reactive Sn and O atoms with low coordination numbers [399]. These surface atoms could serve as centers to efficiently adsorb gaseous molecules, and/or as catalytically active sites, favoring the surface reactions with participation of adsorbed species.

It should be noted, however, that the proposed technologies very often do not agree well with the large-scale production of gas sensors. For example, in [371,379,385], NiO, Co_3_O_4_, and CuO nanosheets were synthesized directly on an Al_2_O_3_ tube, into which a heater was subsequently mounted inside, and in [363,388], ZnO and NiO nanosheets were synthesized directly on a substrate with already formed contacts (see Figure 40a). Li et al. [371] believed that sensors with nano-sheets grown directly on substrates should have excellent performance, since during the process of synthesis the nanosheets form a network porous nanoscale system that promotes fast diffusion and efficient adsorption of gas over the entire sensor surface, which should significantly improve its sensitivity and reduce the response time. SEM images shown in Figure 40b,c demonstrate that the structure of the gas sensitive layer formed in this way really compares favorably with the structure of the gas sensitive layer formed using conventional thick film technology [364,366]. SEM images on nanosheets used for fabrication of gas sensing layers using thick-film technology are shown in Figure 41. 2D nanomaterials aggregated into three-dimensional (3D) architectures are indeed characterized by a significantly greater porosity and better gas permeability. Therefore, in most of the studies, using 2D nanosheets as active sensing elements, these 2D nanomaterials are aggregated in various porous 3D architectures [363,400]. The nanosheets may self-assemble spontaneously or can grow on pre-existing cores. The 2D nanosheets in 3D assemblies generally require a thickness of several nm or more in order to have sufficient mechanical strength to sustain an open architecture under dry conditions [400]. However, in this article we will not analyze gas sensors developed on the basis of such porous 3D assemblies. A special article (the second part of our review) will be devoted to the consideration of such 3D nanomaterials, mesoporous-macroporous, and hierarchical structures.

As for the porosity or gas permeability of 2D-based nanomaterials used in the formation of a gas-sensitive layer by the methods of thick-film technology, several approaches have been proposed to solve this problem. In particular, Cao et al. [384] for this purpose used surface decoration of In_2_O_3_ nanosheets (NSs) by WO_3_ nanoparticles. As a result, both an increase in sensitivity and an improvement in the rate of sensor response were achieved. It can be assumed that WO_3_ nanoparticles provide the necessary gap between In_2_O_3_ nanosheets. Choi et al. [388] used another approach. For the effective penetration of gas molecules into the sensing layers, sub-5 nm-scale pores were generated on the Ru oxide NSs by an ultrafast optical sintering technique. Wang et al. [372] showed that nanopores with a diameter of ~ 3 nm in ultrathin In_2_O_3_ nanosheets with a thickness of 3.5 nm can also be formed in the process of their synthesis. Wang et al. [372] suggested that the formation of mesopores in In_2_O_3_ nanosheets should be attributed to the removal of glycerol ligands during the hydrolysis process. Huang et al. [401] showed that ZnO nanosheets, fabricated by an annealing monodisperse basic zinc nitrate (BZN) nanosheets at 300 °C for 2 h in air, are also highly porous (see Figure 42).

In addition, in many works devoted to the development of gas sensors based on 2D nanomaterials, the temperatures above 200 °C were not used [372,374,380]. Moreover, in some works [373,379,387] an additional heat treatment was not used at all. As a rule, this decision was due to the fact that the heat treatment at an elevated temperature led to a drop in the sensor response (see Figure 43). This suggests that one of the reasons for the increased response of gas sensors based on 2D nanomaterials may be the small contact area between the nanosheets, due to the feature of the gas-sensitive layer structure (see Figure 44). The absence of high temperature treatments suggests that we do not have any sintering between nanosheets, and, therefore, usual mechanical contacts are formed between nanosheets. The last one also means that the adhesion of the formed layers to the substrate and the stability of inter-nanosheets contacts may not be good enough for a long-term operation in various conditions. As a rule, only 2D nanomaterials-based layers formed by methods of thick-film technology were subjected to heat treatments at a temperature of 350–600 °C [200,369,375,378,388].

We must not forget the fact that the smaller the thickness of nanosheets, the worse their stability. Wang et al. [368] drew attention to this in their research. Studying the synthesis of 2D WO_3_ nanosheets with a wall thickness of ~10 nm and analyzing the parameters of gas sensors, they found that the stability of much thinner WO_3_ nanosheets is poorer. In this case, the removal of templates P123 (polyethylene oxide-polypropylene oxide-polyethilene oxide) resulted in the formation of large agglomerates, accompanied by a decrease in the sensor response. Zhang et al. [375] also pointed to the importance of this process. They noted that gas sensors with highly agglomerated ZnO nanosheets had significantly worse parameters. This is natural, since the existence of channels and gaps between the nanosheets is a necessary condition for providing good gas permeability of the sensitive layer required to achieve a high sensitivity and fast response of gas sensors [74].

#### 3.4.3. Sensors on Individual 2D Nanomaterials

It seems that many of the problems described earlier can be solved through the manufacture of sensors based on individual 2D nanomaterials, since with this approach it is possible to fully realize the advantages of 2D nanomaterials. However, the experience of using 1D nanomaterials shows that in this way there are many technological difficulties. Unfortunately, individual nanosheets or nanoflakes, as it was observed earlier with the 1D nanomaterials, are difficult to handle. Therefore, for the transfer of individual nanosheets, it is necessary to develop special technologies and instruments that are poorly compatible with the technologies intended for the mass production of gas sensors [403]. Usually for the transfer of layered 2D materials onto different substrates at a desired location it is necessary to use a long working distance optical inspection system in combination with an XYZθ direction micro-manipulator for an accurate placement, and a certain polymer layer as a transfer medium. The PMMA/sacrifice layer, PDMS, thermoplastic polymer, and hybrid stamp composed of PDMS/PPC or (PC, PMMA)/hBN are usually used as polymer carriers [403]. The comparison of the methods designed for the 2D materials transfer is given in Table 5. The transfer process involves several steps (see Figure 45). At the first stage, the polymer carrier is spin-coated onto a Si/SiO2 substrate or a glass slide and then thin flakes of 2D nanostructures are mechanically exfoliated onto the carrier. In the second stage: The substrate with a polymer layer and 2D nanomaterial is attached on the micro-manipulator and transferred to the desired location. At the final stage, the substrate-carrier rises slowly and the residual polymer in the acetone solution is removed. For the transfer media, such as a thermoplastic polymer and a hybrid stamp, additional heating of the substrates is required to release the stacking. Given the complexity and labor input of this process, atomically thin nanoflakes and nanosheets, were mostly employed for sensing in a suspended form to detect solutes [400].

It should be recognized, however, that there are works aimed at the formation of 2D metal oxide thin films, as close as possible to the monolayer. In particular, a few studies focused on the formation of a monolayer film of 2D metal oxides were conducted by Sasaki’s team [404,405]. They successfully fabricated a closely packed monolayer film of 2D titania nanosheet by the electrostatic self-assembly method [404] and Langmuir−Blodgett technique [405]. Titania nanosheets, used in the last experiments, were 10–30 µm in lateral size and 1.2–1.3 nm in thickness. A cross-sectional TEM image of the 10-layer film of Ti_0.87_O_2_ nanosheets deposited on the atomically flat substrate of SrRuO_3_ revealed that the substrate surface is uniformly covered with the ultrathin film (Figure 46). However, this study was not focused on the application of monolayer films of 2D nanosheets in gas sensors. Therefore, the studies performed by [406] can be considered as a first paper devoted to the fabrication of chemiresistive gas sensors with a monolayer film of 2D nanomaterial. For these purposes Miao et al. [406] used copper(II) oxide nanosheets, synthesized using the hydrothermal method (Figure 47a). Nanosheets had a length and width ranging from 300 to 1000 nm, and the thickness was around 30 to 50 nm (Figure 47b). A monolayer film of copper(II) oxide nanosheets (Figure 47c) was prepared by the self-assembly method at the air−water interface (see Figure 48). After the formation of a monolayer at the air−water interface, the lab jack was raised up at a rate of 5 mm/min until the glass holder was above water level to transfer the monolayer onto the substrates. Then, films were dried at room temperature for 30 min. The as-prepared monolayer with contacts was annealed in a furnace at 400 °C for 6 h before testing of gas sensing characteristics. Miao et al. [406] showed that relying on this method and using the layer-by layer deposition approach, one can also form a multilayer film, based on different 2D nanomaterials.

Miao et al. [406] stated that this method provides a facile, versatile, and highly reproducible manufacturing of chemiresistive gas sensors with the sensing layer based on a monolayer film of 2D nanomaterials. However, this method of manufacturing chemiresistive gas sensors is not a method intended for large-scale production. The main result of the research conducted by Miao et al. [406], is a conclusion that an increase in the number of nanosheets in gas sensitive layers, as previously predicted, is accompanied by a decrease in sensitivity (see Figure 49a) and an increase in the response time. The negative effect of the thickness of the sensing film is explained by the effect of dilution of the analyzed gas and impeded gas diffusion (Figure 49b), which increase with an increasing thickness of a sensing layer.

## 4. Summary

As is known, for the large-scale implementation of new materials and technologies, it is necessary that devices based on them are much cheaper and have significantly better parameters compared to the samples, existing on the market. Unfortunately, the materials that were considered in this article do not provide a significant improvement in the parameters of gas sensors. Despite the stated advantages of 1D and 2D nanomaterials for application in gas sensors, these sensors are far from perfect. Moreover, the technologies used in the manufacture of gas sensors based on these nanomaterials are more expensive than those used in the fabrication of conventional metal oxide-based gas sensors. In addition, many of these technologies are hard-compatible with the processes used in large-scale production of gas sensors.

Of course, over the past decade, various fabrication and characterization strategies have been developed with the aim to accomplish electrical measurements of individual 1D and 2D metal oxide structures free of parasitic effects, and to develop gas sensors that are competitive in parameters with conventional devices. For instance, low-current measurement protocols were proposed that allowed devices based on 1D nanomaterials to operate for a long time without degradation of their performance [158,159,235]. Thus, it can be expected that the development of new technologies adapted to nanomaterials can lead to complete and well-controlled characterization of devices based on 1D and 2D structures, which were previously unattainable [5]. However, despite the significant successes achieved in the development of new 1D and 2D nanomaterials, the improvement of their synthesis technologies, and the manufacture of gas sensors based on them, do not expect the appearance on the sensor market of gas sensors based on individual 1D and 2D nanomaterials. This means that with traditional gas sensors developed based on the nanocrystalline metal oxides using conventional well-developed thin-film and thick-film technology, a long time will dominate the sensor market [24,29,74,97].

Regarding 1D nanostructures arrays and 2D nanostructures assembled into three-dimensional (3D) architectures such as nanoflowers, hollow spheres, etc., we must state that these nanostructures have much more prospects for appearance in devices developed for the sensor market. After appropriate optimization of the technology of 1D and 2D nanomaterials synthesis, which will allow both to form nanostructures with controlled and reproducible parameters, and to improve the sensors performances, sensors based on 1D nanostructures arrays and assembled 2D nanomaterials may well compete with gas sensors based on conventional metal oxides. A detailed discussion of assembled 1D and 2D nanomaterials and gas sensors, developed on their basis, will be presented in the second part of this article.

## Figures and Tables

**Figure 1 nanomaterials-10-01392-f001:**
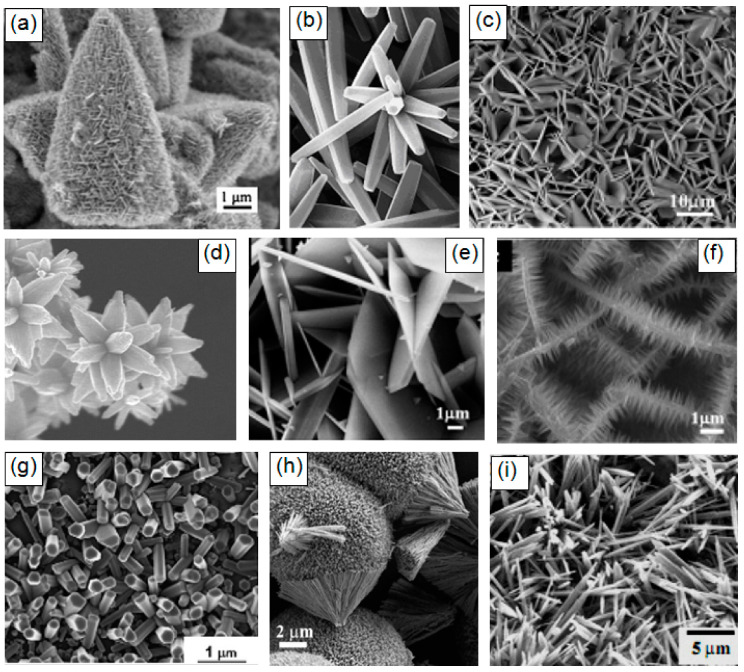
SEM images of ZnO and Cu_2_O nanostructures synthesized by different methods. Reproduced with permission from (**a**) Lu et al. [87]. Copyright 2008: Willey-VCH Verlag GmbH; (**b**) Li et al. [88]. Copyright 2008: Elsevier; (**c**,**e**,**f**) Xu et al. [89]. Copyright 2007: ACS; (**d**) Sepulveda-Guzman et al. [90]. Copyright 2009: Elsevier; (**g**) Krunks et al. [91]. Copyright 2006: Elsevier; (**h**) Orel et al. [92]. Copyright 2007: ACS; (**i**) Dev et al. [93]. Copyright 2006: Institute of Physics.

**Figure 2 nanomaterials-10-01392-f002:**
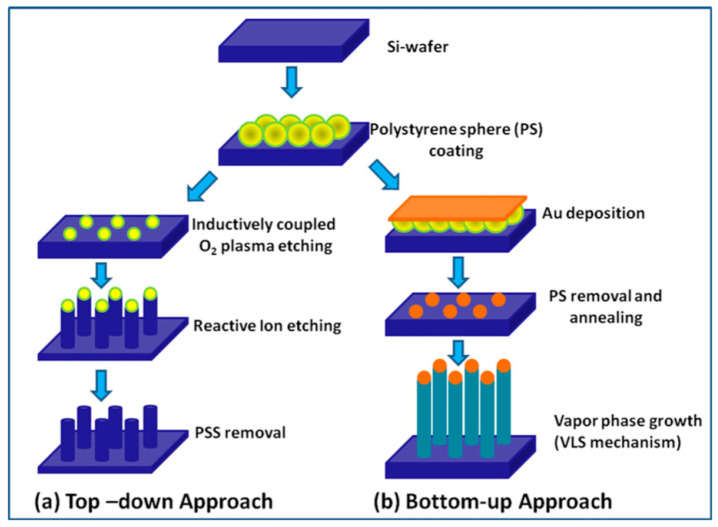
Schematic presentation of general steps involved in the fabrication of nanowires (NWs) using (**a**) top-down and (**b**) bottom-up approaches. Reproduced with permission from Ramgir et al. [130]. Copyright 2013: Elsevier.

**Figure 3 nanomaterials-10-01392-f003:**
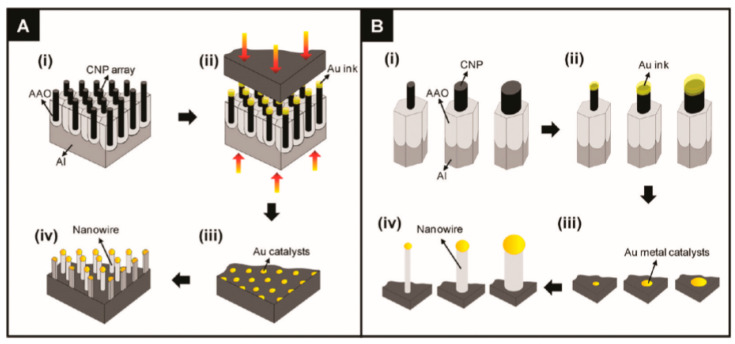
The schematic diagrams depict (**A**) the entire procedure, which includes (i) the fabrication of the carbon nanopost (CNP) stamps, (ii) the contact-printing of the Au ink, (iii) the preparation of the Au metallic catalysts, and (iv) the growth of the semiconductor NWs over the contact-printed Au catalysts via the vapor-liquid-solid (VLS) process; and (**B**) the conceptual model of the size-controlled fabrication of semiconductor NWs via the contact-printing process using the CNP stamps with different tip diameters: (i) Different-sized CNP tips, (ii) Au ink that was loaded onto the CNP tips, (iii) contact-printed Au catalysts from the CNP tips, and (iv) NWs grown over the size-controlled Au catalysts. The variation of the CNP diameter, which was used as the stamps for contact-printing of the Au catalysts, directly controls the diameter of the NWs that are grown over the printed Au particles. Reproduced with permission from Lee et al. [135]. Copyright 2010: American Chemical Society.

**Figure 4 nanomaterials-10-01392-f004:**
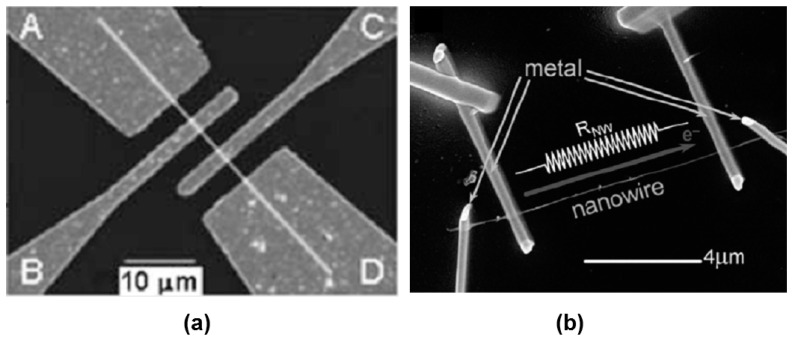
SEM images of gas sensors based on single metal oxide nanobelts and nanowires. (**a**) Reprinted with permission from Fields et al. [148]. Copyright 2006: American Institute of Physics. (**b**) Reproduced with permission from Hernandez-Ramirez et al. [149]. Copyright 2009: Royal Society of Chemistry.

**Figure 5 nanomaterials-10-01392-f005:**
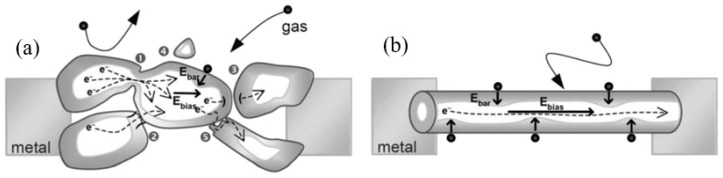
Diagrams illustrating the difference in gas sensing effects in (**a**) the polycrystalline material and (**b**) individual nanowires: One can see that any intergrain necks or boundaries are absent in 1D-based sensors. Moreover, *E*_bar_ and *E*_bias_ fields are always orthogonal and independent. Reproduced with permission from Hernandez-Ramirez et al. [149]. Copyright 2009: Royal Society of Chemistry.

**Figure 6 nanomaterials-10-01392-f006:**
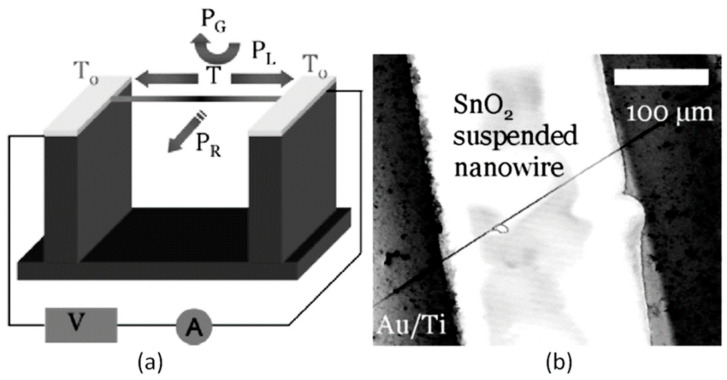
(**a**) The sensor setup and principal thermal losses in the suspended nanowire heated by the Joule heat: *P*_L_-losses to the contacts, *P*_G_- losses to the ambient gas, and *P*_R_ -radiation losses; (**b**) SEM image of the suspended SnO_2_ chemiresistor. Reproduced with permission from Strelcov et al. [168]. Copyright 2009: American Institute of Physics.

**Figure 7 nanomaterials-10-01392-f007:**
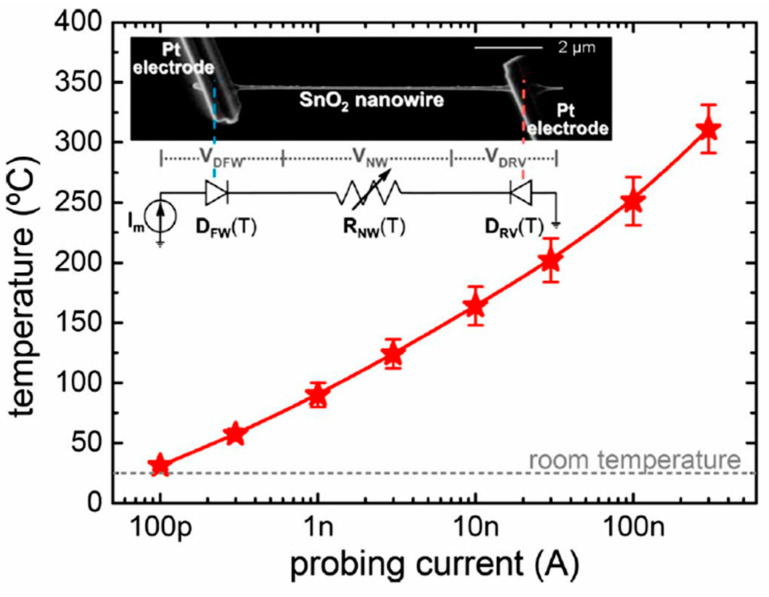
Estimated temperature of the devices at different *I_m_*. (*r_nw_*~35 nm) (inset) SEM image of a SnO_2_ nanowire connected to two Pt microelectrodes fabricated with a focused ion beam. The equivalent circuit of this structure corresponds to two back-to-back diodes in a series with the nanowire resistance. These three components dissipate electrical power and contribute to the self-heating of the device. Reproduced with permission from Prades et al. [169]. Copyright 2008: American Institute of Physics.

**Figure 8 nanomaterials-10-01392-f008:**
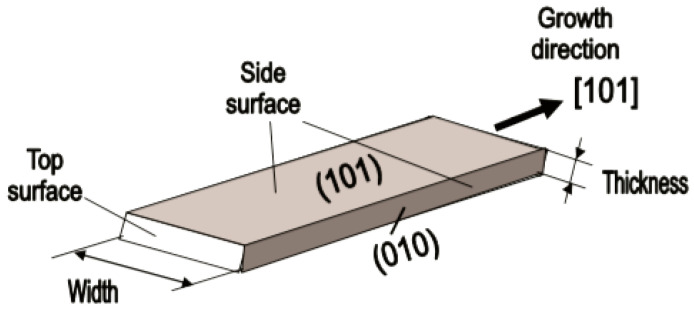
Schematic diagram of the geometric configuration of SnO_2_ nanobelts. Reproduced with permission from Korotcenkov [74]. Copyright 2008: Elsevier.

**Figure 9 nanomaterials-10-01392-f009:**
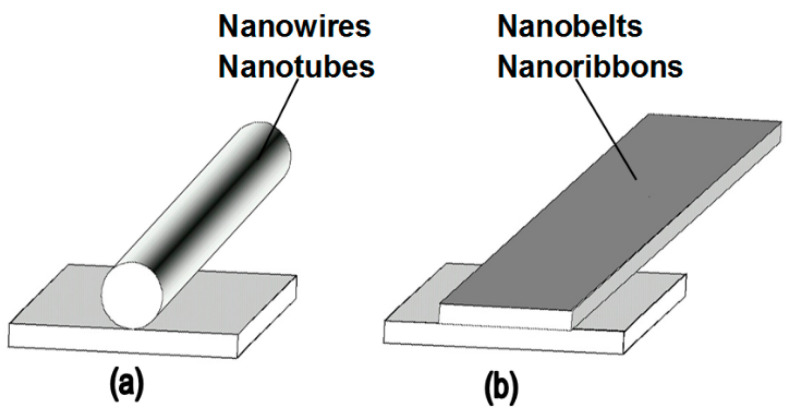
Position of (**a**) nanowires or nanotubes and (**b**) nanobelts on the contact pad. Reproduced with permission from Korotcenkov [74]. Copyright 2008: Elsevier.

**Figure 10 nanomaterials-10-01392-f010:**
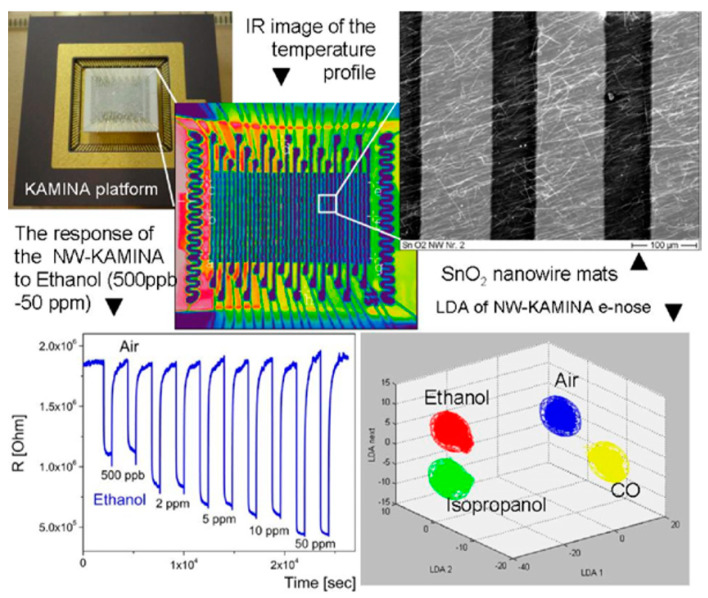
A demonstration of the KAMINA e-nose system performance with SnO_2_ nanowires as a sensing media. Reproduced with permission from Sysoev et al. [213]. Copyright 2007: American Chemical Society.

**Figure 11 nanomaterials-10-01392-f011:**
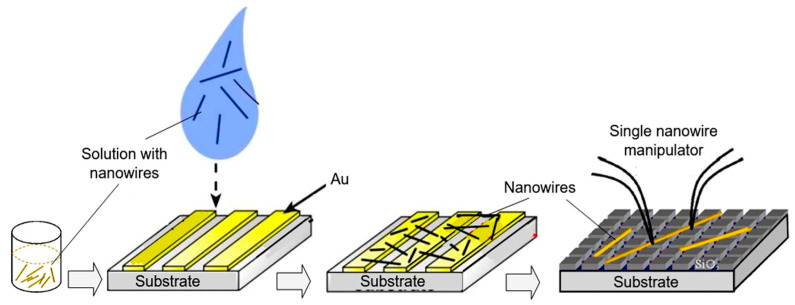
Illustration of the process of transferring 1D structures to the surface of the substrate: Nanowires are removed into a suspension of the solvent. A drop of the nanowire solution is dispersed on a template substrate and evaporated under a vacuum. Finally, nanowires with manipulator tips can be moved from a template substrate to the right place.

**Figure 12 nanomaterials-10-01392-f012:**
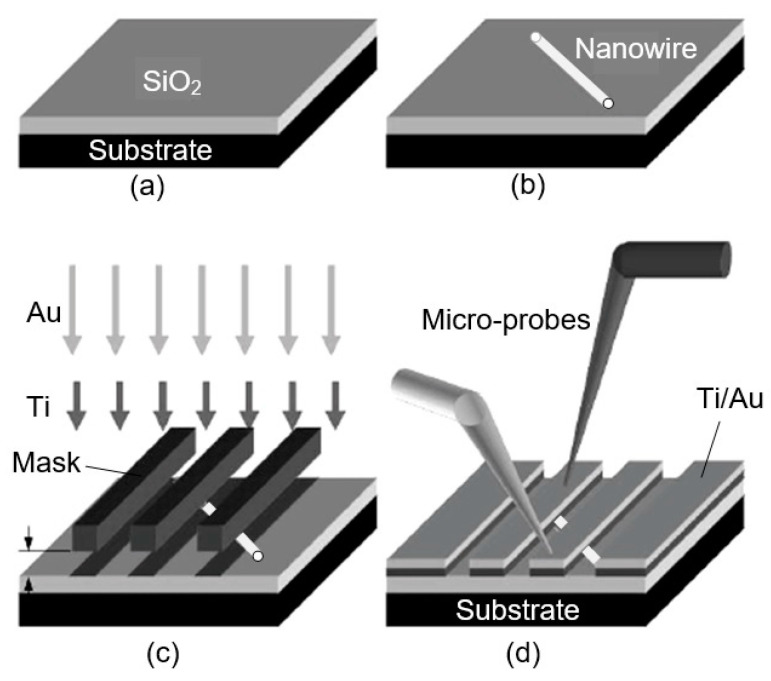
Protocol for a resist-free fabrication of 1D metal oxide nanowire-based chemiresistors and chemFETS: (**a**) Pristine Si/SiO_2_ wafer; (**b**) nanostructures placed on wafer mechanically; (**c**) shadow masking to determine metal contacts; (**d**) microprobes can be used to explore transport and sensing properties of the individual nanoresistors. Idea from [234].

**Figure 13 nanomaterials-10-01392-f013:**
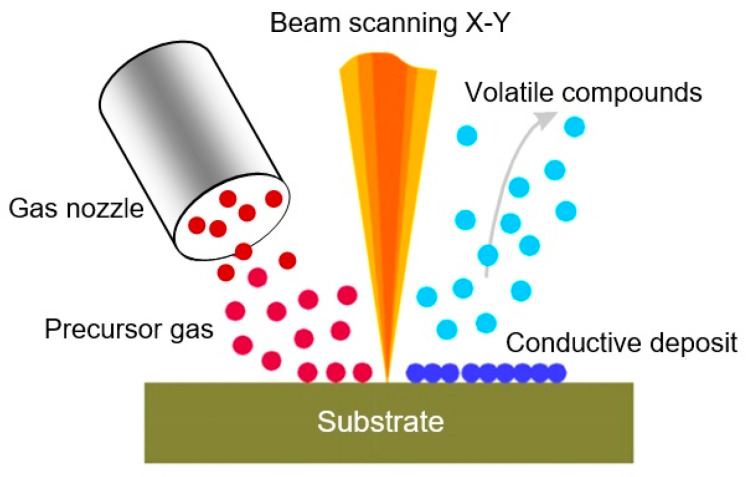
Schematic representation of local deposition, assisted by the focused ion beam (FIB). The gas precursor introduced by the capillary is decomposed by the secondary electrons generated by the interaction of the primary ion beam with the target. Reproduced with permission from Gierak [237]. Copyright 2009: Institute of Physics.

**Figure 14 nanomaterials-10-01392-f014:**
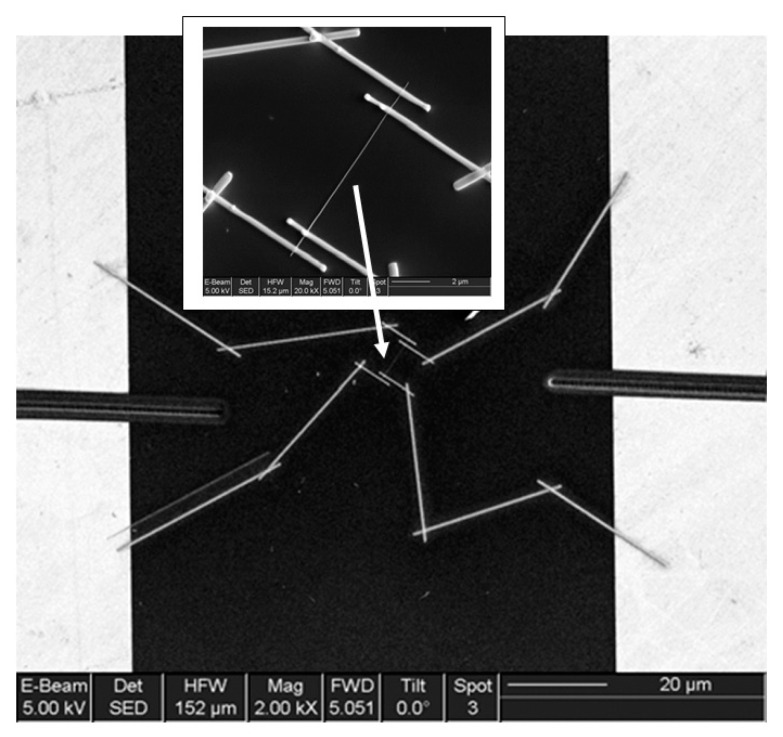
General view of the final device based on the SnO_2_ individual nanowire (dimensions: L = 11 μm (length) and D = 55 ± 5 nm (diameter)) with Au/Ti/Ni microelectrodes. The position of the contacted NW is indicated by the arrow. Reproduced with permission from Hernandez-Ramirez et al. [159]. Copyright 2007: Elsevier.

**Figure 15 nanomaterials-10-01392-f015:**
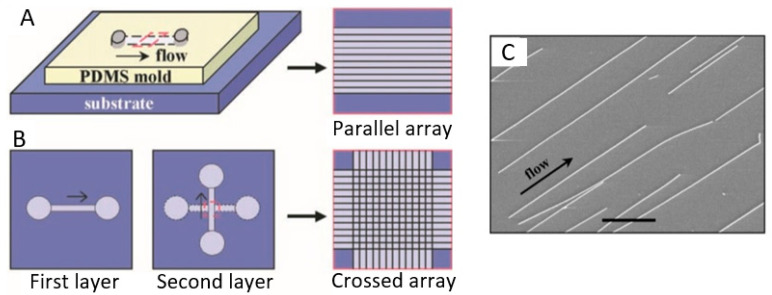
Schematic of fluidic channel structures for flow assembly: (**A**) A channel formed when the polydimethylsiloxane (PDMS) mold was brought in contact with a flat substrate. The NW assembly was carried out by flowing a NW suspension inside the channel with the controlled flow rate for a set duration. Parallel arrays of NWs are observed in the flow direction on the substrate when the PDMS mold is removed. (**B**) The multiple crossed NW array can be obtained by changing the flow direction sequentially in a layer-by-layer assembly process. (**C**) SEM image of the parallel assembly of NW arrays formed using the micro fluidics method. Reproduced with permission from Huang et al. [270]. Copyright 2001: AAAS.

**Figure 16 nanomaterials-10-01392-f016:**
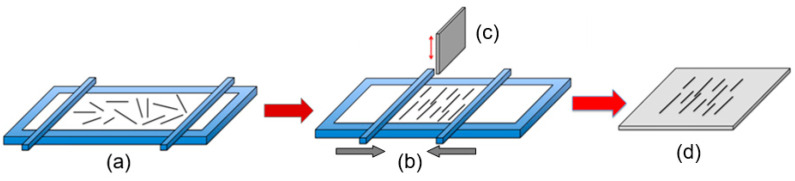
Schematic diagram of the Langmuir-Blodgett technique for NWs alignment. (**a**) Random nanowires suspended in the Langmuir-Blodgett trough; (**b**) monolayer compression; (**c**) wafer being pulled vertically from the suspension in parallel with the lateral motion of the barrier; (**d**) resulting parallel nanowire array on the substrate.

**Figure 17 nanomaterials-10-01392-f017:**
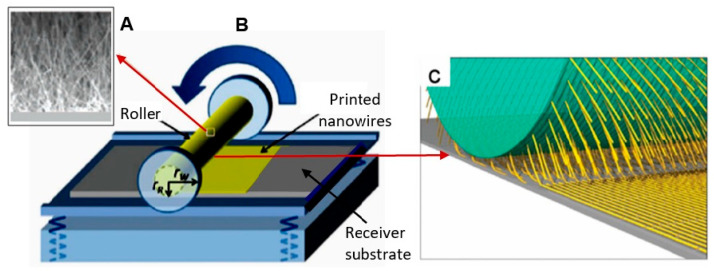
Schematic diagram illustrating the roll printing technique for NWs alignment: (**A**) SEM image of NWs grown perpendicularly to the surface of the cylindrical substrate; (**B**) installation for NWs alignment; (**C**) the mechanism of a nanowire transfer. The NWs are oriented and transferred to the receiving substrate by applying a directional shear force, resulting in the printing of sub-monolayer parallel NW arrays on the receiving substrate. Reproduced with permission from Hu et al. [269]. Copyright 2020: Royal Society of Chemistry.

**Figure 18 nanomaterials-10-01392-f018:**
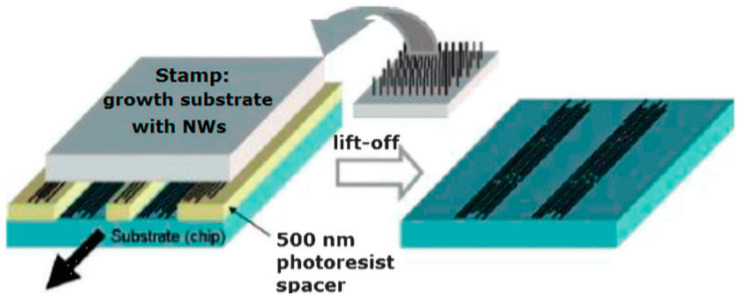
Contact printing of NWs from a growth substrate on a prepatterned substrate. In general, NWs are grown perpendicularly to substrate with a random orientation, but they can be well-aligned by shear forces during the printing process. Reproduced with permission from Javey et al. [271]. Copyright 2007: Royal Society of Chemistry.

**Figure 19 nanomaterials-10-01392-f019:**
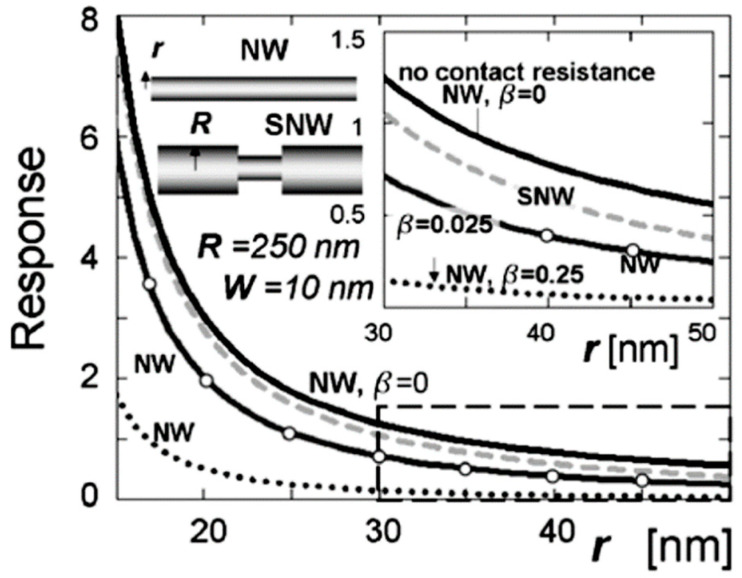
The response of straight and segmented nanowires as a function of their radius at various contact resistances (*b* = 0, 0.025, 0.25). For segmented nanowires (SNWs), the curve is drawn versus the radius of the smaller segment. The solid curve (top) corresponds to the nanowire with no contact resistance; the dashed curve corresponds to the SNW with thick segments of a 500 nm diameter and *b* = 0.025; the solid curve marked with circles corresponds to the SNW with *b* = 0.025; the dotted curve corresponds to the SNW with *b* = 0.25. The depletion width is ~10 nm at all cases. Reproduced with permission from Dmitriev et al. [274]. Copyright 2007: Institute of Physics.

**Figure 20 nanomaterials-10-01392-f020:**
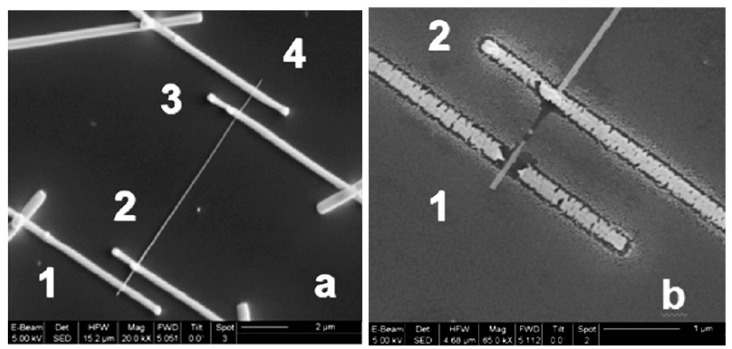
(**a**) A fabricated FIB four-probes contacted SnO_2_ nanowire and (**b**) a detail of the contact after an electrical failure likely caused by the excess of heat dissipation in the contacts. Reproduced with permission from Hernandez-Ramirez et al. [159]. Copyright 2007: Elsevier.

**Figure 21 nanomaterials-10-01392-f021:**
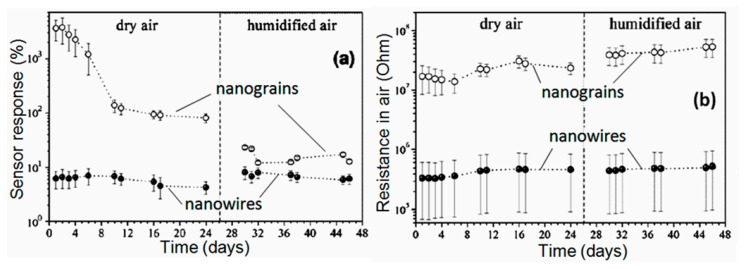
The change of sensing characteristics of the median sensor segment of a SnO_2_ 3D mesoporous nanoparticle (NP) or nanograins (NG) layer and nanowire (NW) mat versus measurement day: (**a**) Sensitivity or response to 1 ppm of 2-propanol vapors; (**b**) background resistance in air. Open and filled circles correspond to the NP and NW samples, respectively. Reproduced with permission from Sysoev et al. [129]. Copyright 2009: Elsevier.

**Figure 22 nanomaterials-10-01392-f022:**
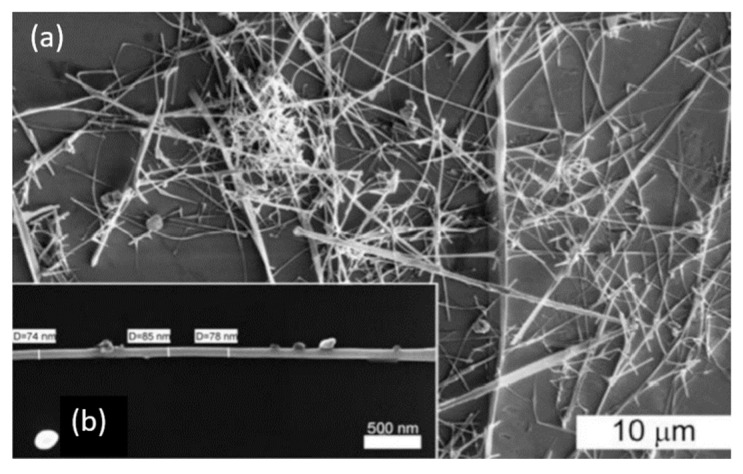
(**a**) SEM image of SnO_2_ nanowires after the synthesis using the gas transport method based on the vapor-liquid-solid (VLS) mechanism, and (**b**) SEM image of a single SnO_2_ nanowire. Reproduced from Shaposhnik et al. [292]. Published by Beilstein Sci. as open access.

**Figure 23 nanomaterials-10-01392-f023:**
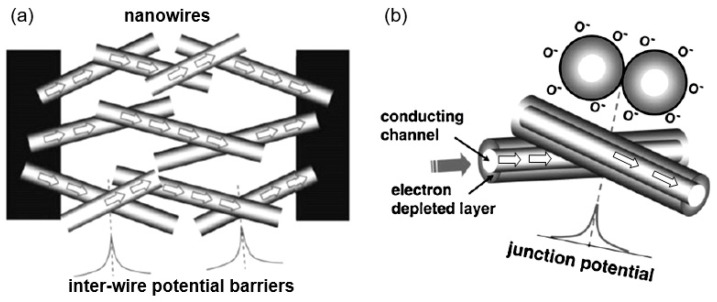
(**a**) Schematic diagrams for multi-nanowire-based chemical sensors; (**b**) schematic illustration of the gas sensing mechanism in a network of nanowires. Reproduced with permission from Vomiero et al. [294]. Copyright 2007: American Chemical Society.

**Figure 24 nanomaterials-10-01392-f024:**
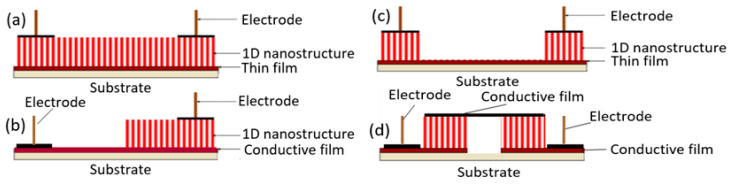
(**a**–**d**) Schematic diagrams for a multi-nanowire-based sensor with nanotubes, nanowires, or nanorods oriented in a perpendicular direction.

**Figure 25 nanomaterials-10-01392-f025:**
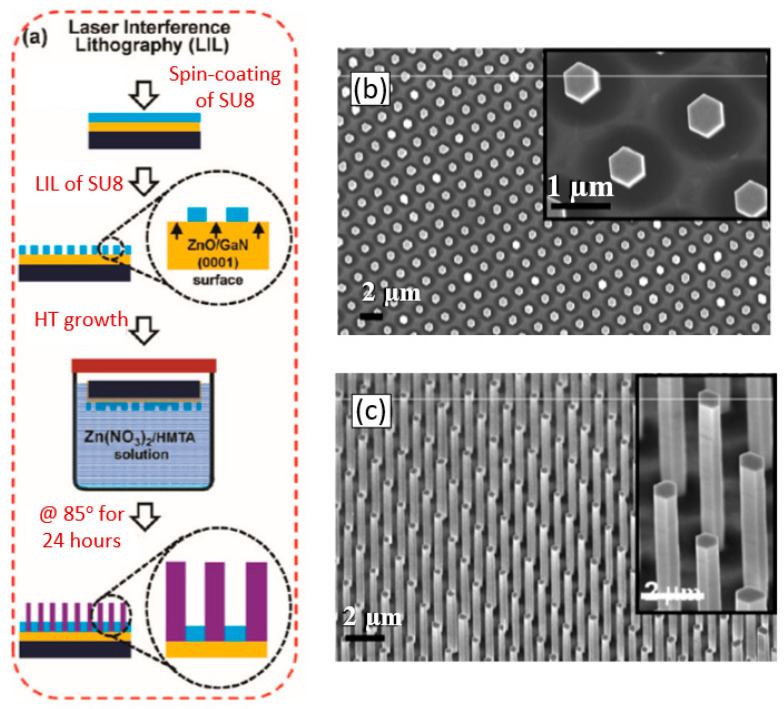
(**a**) Schematics of the fabrication sequences of vertically aligned ZnO NW arrays using laser interference lithography (LIL); (**b**) optical image of a 2 in. Si wafer with a SU-8 open-hole pattern over the whole surface area. The iridescence dispersion demonstrates the excellent periodicity over the entire wafer surface. (**c**) The top-view SEM image of the patterned SU-8 film (thickness of 500 nm). Inset, the top-view SEM image of the patterned SU-8 film at higher magnification. The area encircled by the black dashed line is the exposed surface of the substrate. Reproduced with permission from Wei et al. [302]. Copyright 2010: American Chemical Society.

**Figure 26 nanomaterials-10-01392-f026:**
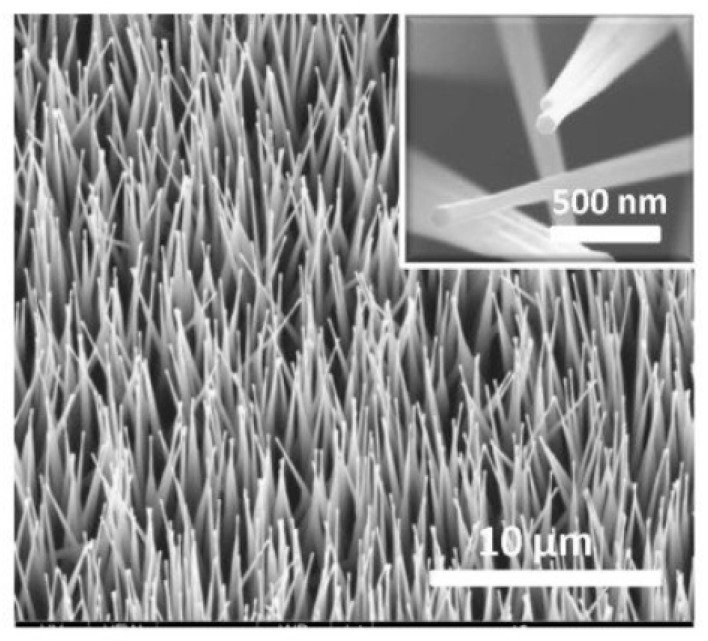
FESEM image (tilted at 15°) of a directly synthesized vertically aligned ZnO nanowire array on a fluorine-doped tin oxide (FTO) substrate. The inset is a higher-magnification image. Reproduced from Lu et al. [306]. Published by Springer as open access.

**Figure 27 nanomaterials-10-01392-f027:**
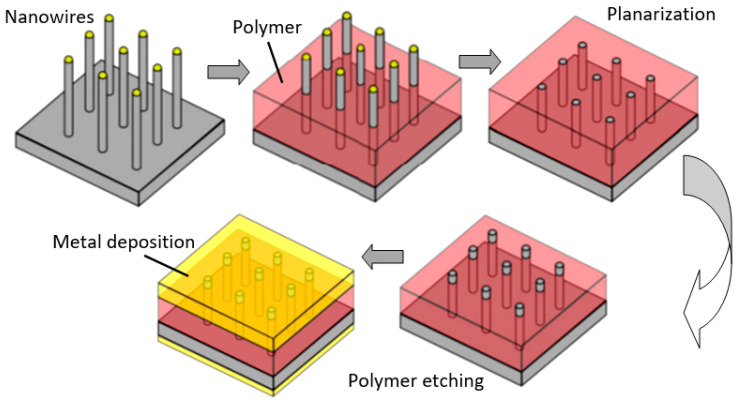
Schematic illustration of the contact forming on the top of vertically grown NWs.

**Figure 28 nanomaterials-10-01392-f028:**
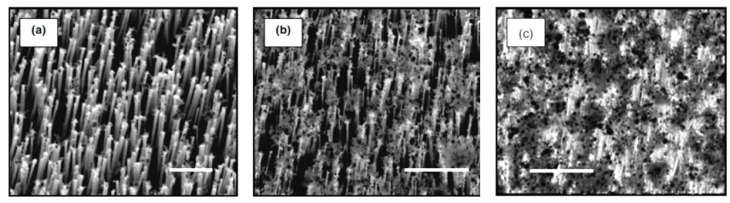
SEM images of Au nanoparticles attached to the tips of the nanowire assembly at different stages of the deposition: (**a**) 10 min; (**b**) 1 h; (**c**) 2 h. Scale bars = 1 µm. Reproduced with permission from Parthangal et al. [309]. Copyright 2007: Cambridge University Press.

**Figure 29 nanomaterials-10-01392-f029:**
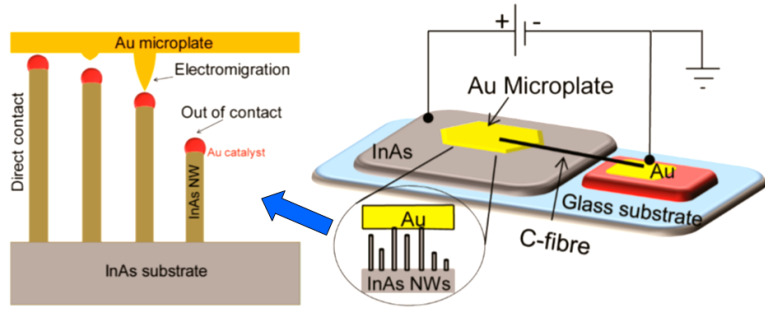
Schematic showing the device configuration of Au microplate/InAs NWs sandwich. Reproduced with permission from Radha et al. [310]. Copyright 2012: American Chemical Society.

**Figure 30 nanomaterials-10-01392-f030:**
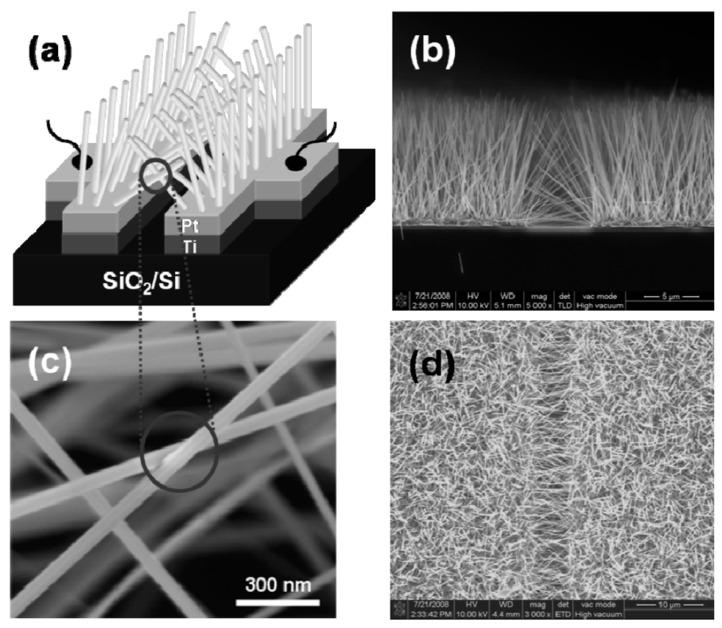
(**a**) Schematic illustration of ZnO-nanowire air bridges over the SiO_2_/Si substrate. (**b**) Side- and (**d**) top-view SEM images clearly show the selective growth of ZnO nanowires on the Ti/Pt electrode. (**c**) The junction between ZnO nanowires, grown on both electrodes. Reproduced with permission from Ahn et al. [312]. Copyright 2009: Elsevier.

**Figure 31 nanomaterials-10-01392-f031:**
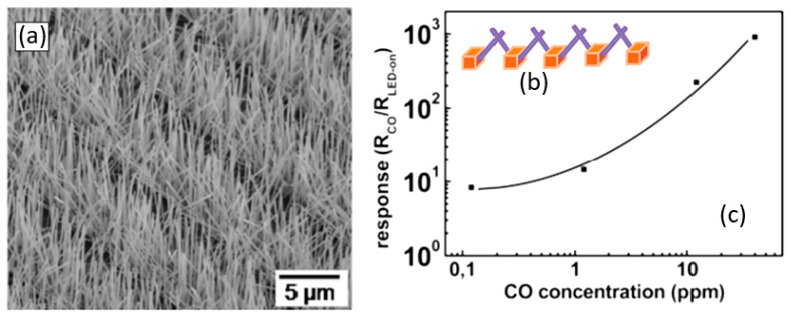
(**a**) SEM images of ZnO NWs grown on substrates with the Au electrodes (electrode line width × gap distance = 3 × 5 µm); (**b**) schematic representation of the nanobridge junctions formed and (**c**) sensor response to the CO gas. Reproduced with permission from Yuon et al. [313]. Copyright 2010: American Chemical Society.

**Figure 32 nanomaterials-10-01392-f032:**
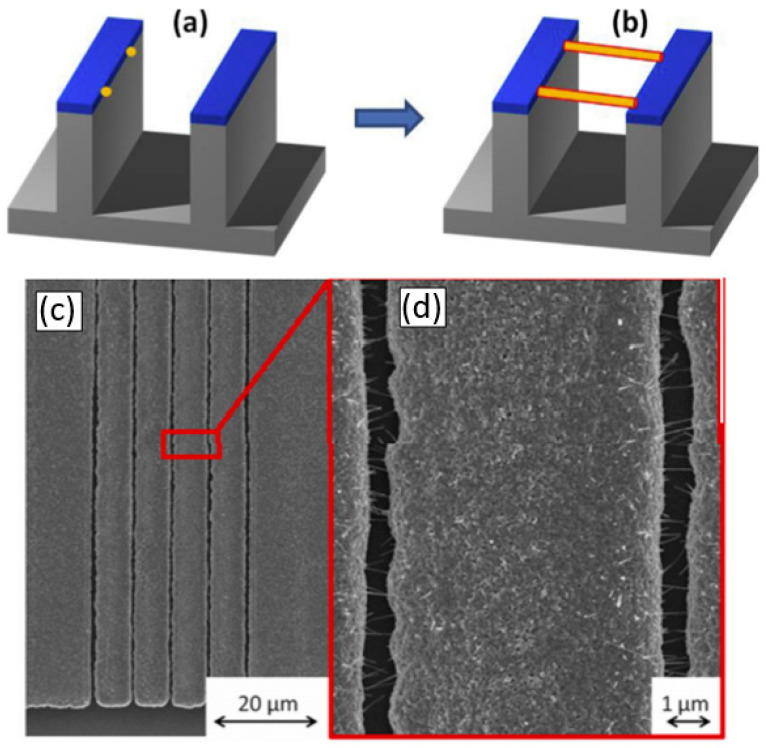
(**a**,**b**) Schematic of the bridging NW growth. Reproduced with permission from Huang et al. [219]. Published by Elsevier as open access; (**c**,**d**) SEM images of the fabricated CuO NWs-based gas sensor: (c) Low magnification; (d) high magnification. Reproduced with permission from Steinhauer et al. [163]. Copyright 2013: Elsevier.

**Figure 33 nanomaterials-10-01392-f033:**
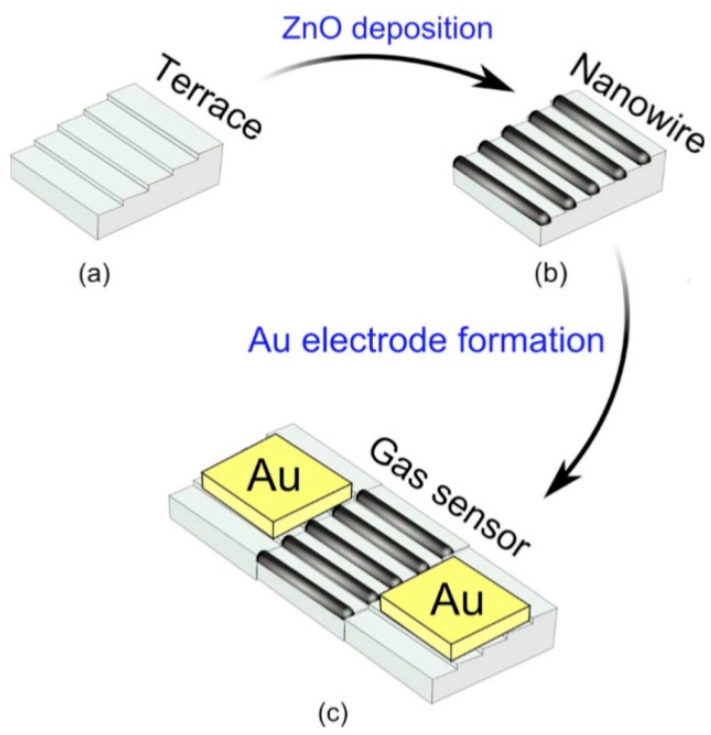
Diagram of the gas sensor fabrication process: (**a**) Uniform terraces formed on the (0001) sapphire substrate, (**b**) nanowires growth on the terrace using the laser pulsed layer deposition method, (**c**) nanowire-based gas sensor on the terrace. Reproduced with permission from Kim and Son [316]. Copyright 2009: Institute of Physics.

**Figure 34 nanomaterials-10-01392-f034:**
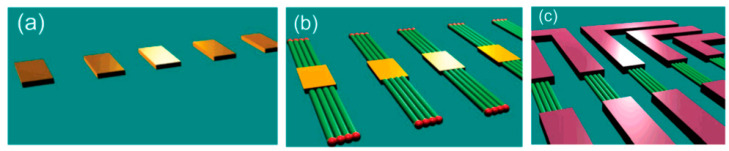
Schematic of the photolithography process for a scalable fabrication of nanowire devices. (**a**) Gold pads and fiducial marks are deposited on the surface. (**b**) NWs are grown selectively from the two sides of the gold pads. (**c**) Metal electrodes and bonding pads are placed exactly on NWs by the alignment of fiducial marks. Reproduced with permission from Nikoobakht [319]. Copyright 2007: American Chemical Society.

**Figure 35 nanomaterials-10-01392-f035:**
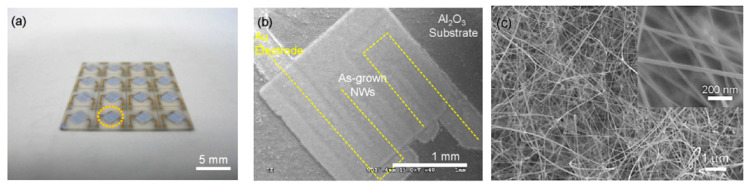
The sensor elements: (**a**) 16 sensor elements prepared by the one-pot growth of SnO_2_ nanowire networks on laser-scriber Al_2_O_3_ substrates; (**b**) SEM image of one sensor element; (**c**) SEM image of SnO_2_ nanowire networks. Reproduced with permission from Hwang et al. [153]. Copyright 2012: Elsevier.

**Figure 36 nanomaterials-10-01392-f036:**
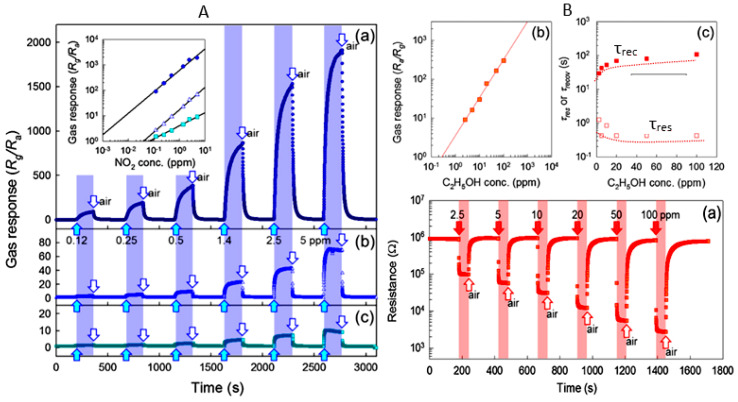
(**A**) Dynamic sensing transients to 0.12–5 ppm NO_2_ at (**a**) 141, (**b**) 190, and (**c**) 240 °C and the corresponding sensor response as a function of NO_2_ concentration (insert); (**B**) (**a**) dynamic sensing transients to 2.5–100 ppm C_2_H_5_OH, (**b**) gas responses as a function of the C_2_H_5_OH concentration, and (**c**) the 90% response times (τ_res_) and 90% recovery times (τ_rec_). Reproduced with permission from Hwang et al. [153]. Copyright 2012: Elsevier.

**Figure 37 nanomaterials-10-01392-f037:**
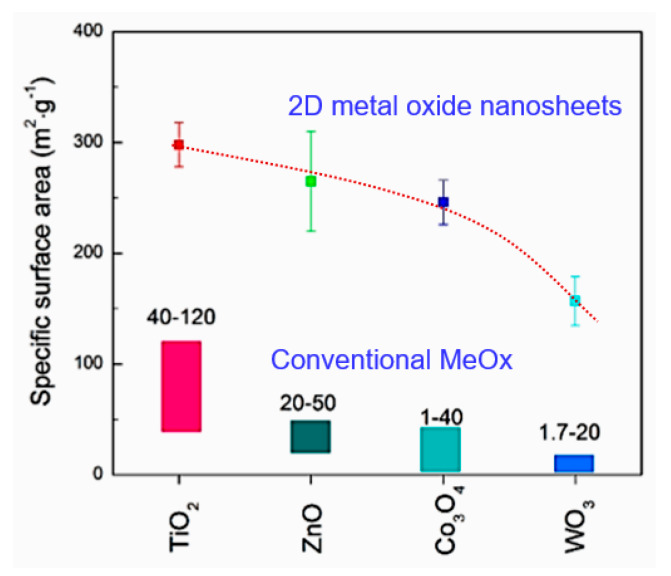
Typical specific surface areas of conventional metal oxide nanoparticles and 2D metal oxide nanosheets. Reproduced from Sun et al. [335]. Published by Nature Research as open access.

**Figure 38 nanomaterials-10-01392-f038:**
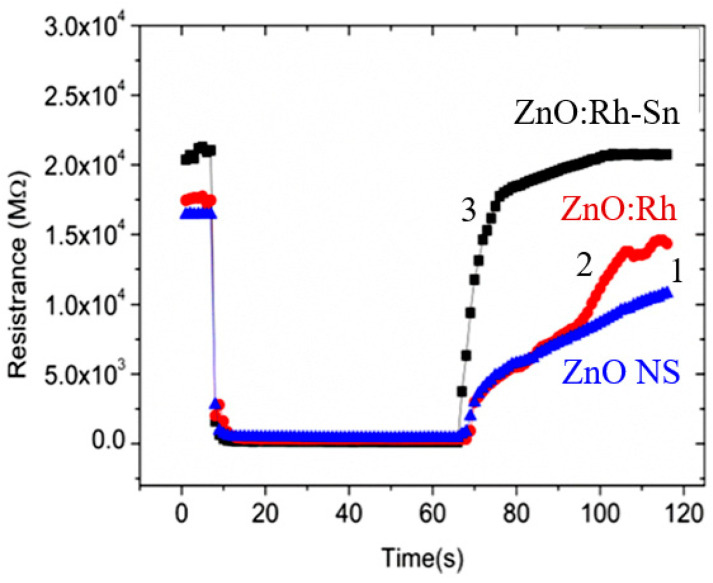
The response and recovery curves of different sensors to 100 ppm ethanol at an optimum working temperature (300 °C): 1—ZnO nanosheets; 2—ZnO:Rh; 3—ZnO:Rh-Sn. Reproduced from Sun et al. [335]. Published by Nature Research as open access.

**Figure 39 nanomaterials-10-01392-f039:**
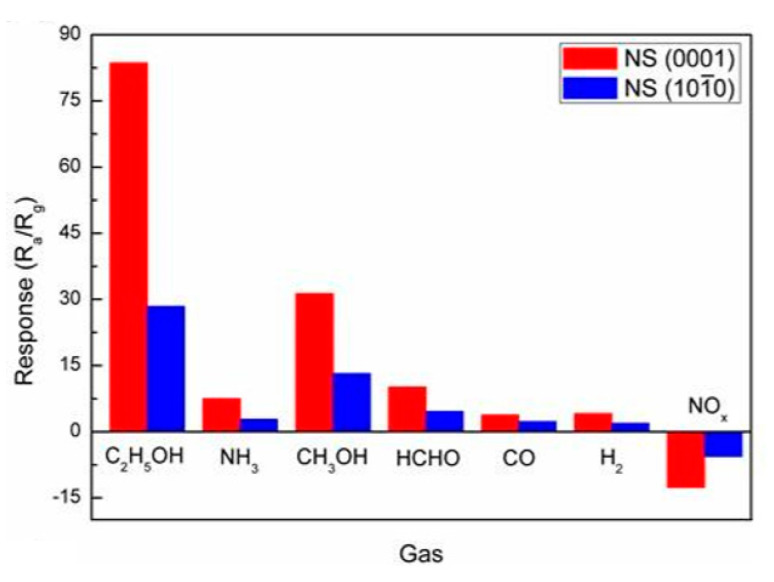
Response of ZnO nanosheets-based sensors to different gases. ZnO nanosheets in these sensors are faceted by different crystallographic planes, (0001) and ± (01$\bar{1}$0), respectively. Reproduced with permission from Xu et al. [200]. Copyright 2017: Elsevier.

**Figure 40 nanomaterials-10-01392-f040:**
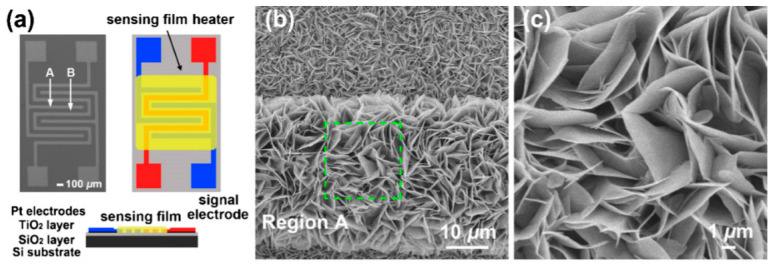
(**a**) Schematic diagrams of the different views for the microstructure sensor. Morphologies and crystal structures of ZnO nanosheets (NSs): (**b**,**c**) FESEM images of ZnO NSs taken from different regions in the microstructure sensor. Reproduced with permission from Zeng et al. [363]. Copyright 2012: Elsevier.

**Figure 41 nanomaterials-10-01392-f041:**
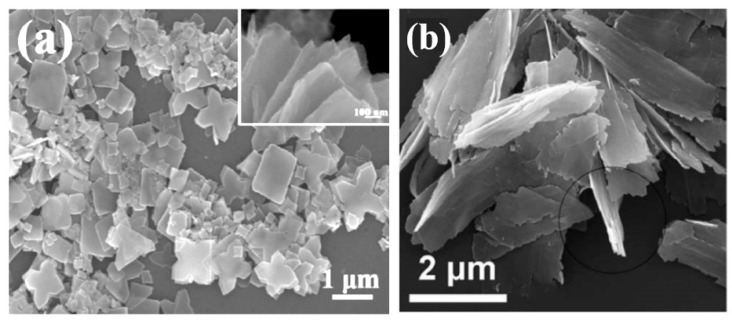
Low and high magnification (inset) FESEM images of WO_3_ (**a**) and Co_3_O_4_ nanosheets (**b**). Nanosheets were synthesized via a hydrothermal method. The thickness of nanosheets was from 20 to 30 nm for WO_3_ and from 20 to 50 nm for Co_3_O_4_ nanosheets. (a) Reproduced with permission from Wang et al. [368]. Copyright 2017: Elsevier; (b) reproduced with permission from Wang et al. [387]. Copyright 2015: Royal Society of Chemistry.

**Figure 42 nanomaterials-10-01392-f042:**
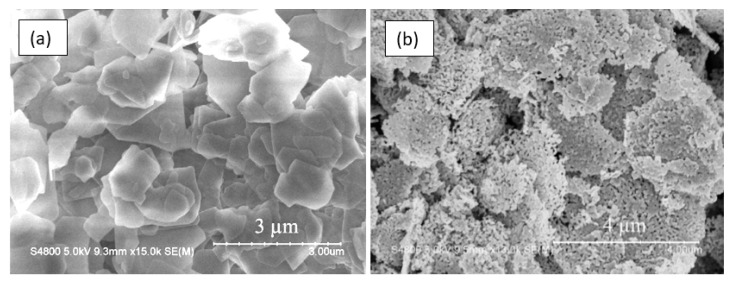
SEM images of (**a**) the as-synthesized BZN nanosheets, and (**b**) porous ZnO nanosheets after BZN nanosheets annealing at 300 °C. The thickness of nanosheets before annealing was in the range of 12–20 nm, and < 15 nm after annealing. Reproduced with permission from Huang et al. [401]. Copyright 2011: Elsevier.

**Figure 43 nanomaterials-10-01392-f043:**
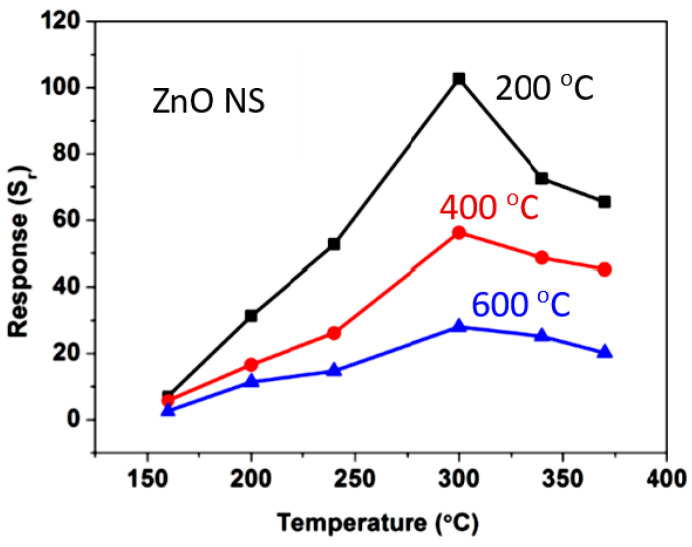
The operating temperature influence on the conductivity response to acetone vapor (200 ppm) of gas sensors based on ZnO nanosheets calcinated at 200, 400, and 600 °C. Reproduced with permission from Li et al. [374]. Copyright 2017: Elsevier.

**Figure 44 nanomaterials-10-01392-f044:**
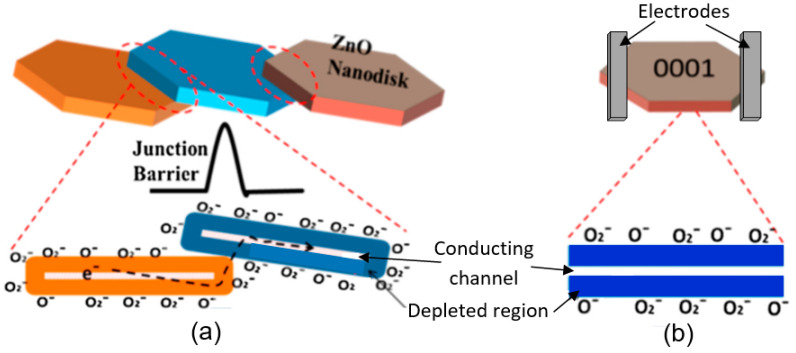
Schematic representation of the sensing mechanism in ZnO gas sensors fabricated using (**a**) a stack of ZnO nanodisks and (**b**) an individual nanodisk. Reproduced with permission from Alenezi et al. [402]. Copyright 2014: American Chemical Society.

**Figure 45 nanomaterials-10-01392-f045:**
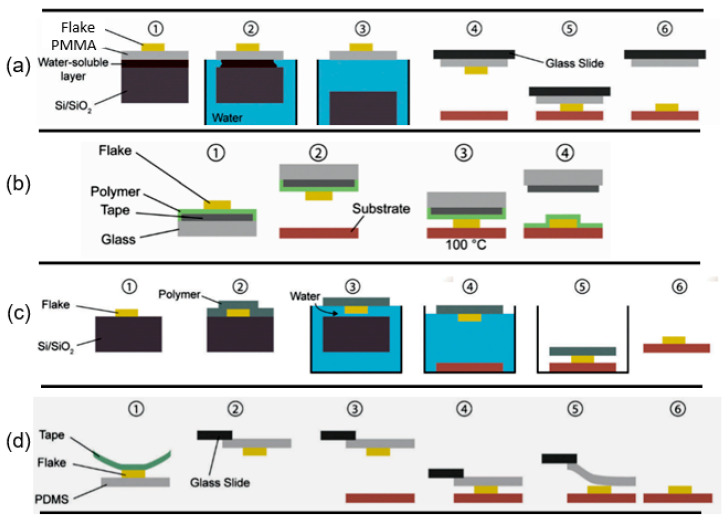
Illustration of transfer methods: (**a**) The PMMA carrying layer transfer method. The designated flake to be transferred is exfoliated onto a Si/SiO_2_ substrate which has been coated with a water-soluble polymer layer and PMMA (1). The stack is then immersed in water (2) where the water-soluble layer can dissolve leaving the PMMA layer carrying the flake floating on the water surface (3). The PMMA is then attached to a glass slide connected to a micromanipulator with the flake facing down (4). With the help of a microscope the flake is aligned with the target substrate and is then brought in contact (5). By gently separating the PMMA from the final substrate the flake gets transferred. (**b**) The Elvacite sacrificial layer transfer method. The target flake is exfoliated onto a layer of Evalcite deposited onto an adhesive tape attached to a glass slide (1). The stack is then positioned onto a target substrate and the flake is aligned with pre-existing features of the substrate (2). The substrate is heated up at 100 °C and the stack can then be brought in contact with the hot substrate (3) which melts the Evalcite layer leaving the capped flake on the final substrate (4). (**c**) The wedging transfer method. The flake to be transferred is exfoliated onto a hydrophilic substrate such as Si/SiO_2_ (1) and then covered by a layer of hydrophobic polymer (2). The stack is then immersed in water where the water molecules can intercalate between the SiO_2_ and the polymer carrying the flake (3). The polymer film carrying the flake remains floating on the water surface where it can be aligned with the help of a needle to the final substrate (4). By pumping down the water the two can be brought in contact (5). Then, the polymer can be dissolved leaving the flake transferred onto the final substrate (6). (**d**) The PDMS dry transfer method. The flake to be transferred is exfoliated onto a PDMS stamp (1) and the stamp is then attached to a glass slide connected to a micromanipulator (2). Using a microscope, the flake can be aligned with the final substrate (3) and brought in contact (4). By slowly peeling the PDMS stamp (5) the flake can be deposited on the substrate (6). Reproduced with permission from Frisenda et al. [403]. Copyright 2018: Royal Society of Chemistry.

**Figure 46 nanomaterials-10-01392-f046:**
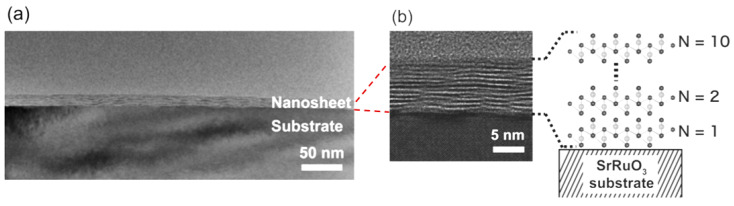
Cross-sectional TEM images of the 10-layer film of Ti_0.87_O_2_ nanosheets formed using the LB deposition technique: (**a**) Wide view; (**b**) magnified image. Reproduced with permission from Akatsuka et al. [405]. Copyright 2009: American Chemical Society.

**Figure 47 nanomaterials-10-01392-f047:**
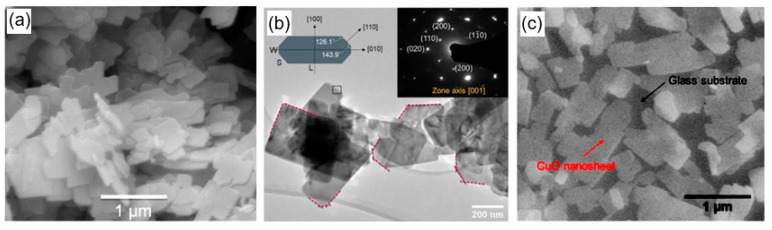
(**a**) SEM images of as-synthesized CuO nanosheets; (**b**) low-magnification TEM image of CuO nanosheets and the SAED pattern and a sketch of the crystal orientation in the real space as inset; (**c**) top surface SEM image of the CuO nanosheet monolayer on the glass substrate. Reproduced from Miao et al. [406]. Published by ACS as open access.

**Figure 48 nanomaterials-10-01392-f048:**
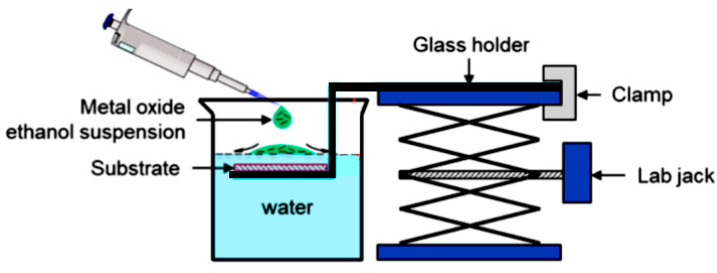
Illustration of the monolayer film formation process using the home-built coater. The red box illustrates that the CuO nanosheets float on the water surface due to the formation of a negative meniscus at the edges and self-assembly due to the attraction force of the same type of meniscus. Reproduced from Miao et al. [406]. Published by ACS as open access.

**Figure 49 nanomaterials-10-01392-f049:**
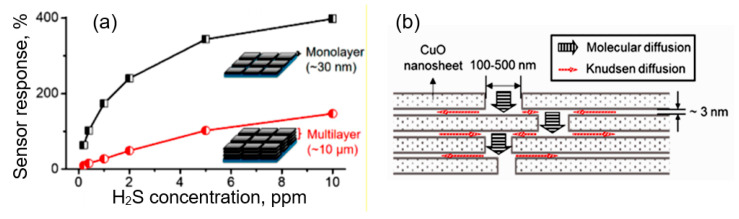
(**a**) Sensor response of monolayer- and multilayer-based sensors to H_2_S; (**b**) model of a multilayer film with lamellar structure. Reproduced from Miao et al. [406]. Published by ACS as open access.

**Table 1 nanomaterials-10-01392-t001:** Several examples of a highly sensitive gas sensor based on one-dimensional (1D) nanostructures.

Gas	Material	D, nm	Sensor Type	T_oper_, °C	DL, ppm	Response	τ_res_, s	Ref.
NO_2_	SnO_2_ (NWs)	35–45	R	25 (UV)	<0.1	2 @ 5 ppm	~60	[152]
				175		~2 @ 10 ppm		
	SnO_2_ (NWs)	30–100	R	240	<0.01	500 @ 1 ppm	~60	[153]
	In_2_O_3_ (NWs)	~10	FET	25	0.5	10^6^ @ 100 ppm	5–600	[154]
	In_2_O_3_ (NWs)	~10	FET	25	0.005	~1.6 @ 200 ppb	50–1000	[155]
H_2_	SnO_2_ (NWs)	50–200	R	300	<5.0	25 @ 50 ppm	<120	[156]
CO	SnO_2_ (NBs)	60	R	400	~5.0	70 @ 250 ppm	30	[157]
	SnO_2_ (SNW)	25–70	R	300	<1.0	2.1 @ 200 ppm	N/A	[158,159]
	ZnO:Pd (NWs)	60–70	R	20	<0.1	1.2 @ 0.1 ppm	>240	[160]
H_2_S	SnO_2_:Cu (NWs)	10–100	R	150	<1.0	~10^7^ @ 80 ppm	N/A	[161]
	In_2_O_3_ (NWs)	30–100	FET	120	<1.0	~12 @ 20 ppm	>300	[162]
	CuO (NWs)	20–40	R	325	~0.01	4 @ 500 ppb	~600	[163]
	ZnO (NWs)	~20	R	300	0.005	~2 @ 200 ppb	~270	[164]
C_2_H_5_OH	ZnO:Mg (NWs)	~120	R	350	~0.01	343 @ 5 ppm	~20 s	[165]
	SnO_2_ (NWs)	30–100	R	300	<1.0	100 @ 20 ppm	<20 s	[153]
NH_3_	WO_3_ (NWs)	~5	R	25	~0.1	1.1 @ 5 ppm	~ 50 s	[166]
	SnO_2_ (SNW)	~100	R	300	~0.1	~2 @ 100 ppm	~ 25 s	[167]

D—diameter; DL—detection limit; FET—field effect transistor; NWs—nanowires; R—resistive; SNW—single NW; T_oper_—operation temperature; Response – R_air_/R_gas_.

**Table 2 nanomaterials-10-01392-t002:** Crystallographic geometry of one-dimensional oxide nanostructures.

Nanostructures	Crystal Structure	Growth Direction	Top Surface	Side Surface
ZnO-belt	Wurtzite	(0001)or (01$\bar{1}$0)	±(2$\bar{1}\bar{1}$0)	±(01$\bar{1}$0) or ±(0001)
Ga_2_O_3_-belt	Monoclinic	(001) or (010)	±(100)	±(010) or ±(10$\bar{1}$)
Ga_2_O_3_-sheet	Monoclinic	(101) (normal)	±(100)	±(010) ±(10$\bar{1}$) and ±(21$\bar{2}$)
t-SnO_2_-belt	Rutile	(101)	±(10$\bar{1}$)	±(010) and ±(10$\bar{1}$)
SnO_2_-belt	Rutile	(100)	±(001)	
t-SnO_2_-wire	Rutile	(101)	±(10$\bar{1}$)	±(010)
SnO_2_-belt (zigzag-initial) (zigzag-final)	Rutile Rutile	(101) (101)	±(010) ±(010)	±(10$\bar{1}$) and ±(100) ±(100)
α-SnO_2_-wire	Orthorhombic	(010)	±(100)	±(001)
SnO_2_-diskette	Tetragonel	±(100) and ±(110)	±(001)	±(100) and ±(110)
SnO_2_-ribbon	Rutile	(101)	(10$\bar{1}$)/($\bar{1}$01)	(010)/(0$\bar{1}0$)
SnO_2_-ribbon (sandwich)	Rutile/orhorom	(110)_o_/(6$\bar{5}$3)_t_	±(100)_o_/±(231)_t_	±(001)_o_/±(10$\bar{1}$)_t_

*Source:* Data from [74,185,201,202,203].

**Table 3 nanomaterials-10-01392-t003:** Summary of the NW assembly technologies.

NW Assembly Technologies	Advantages	Disadvantages
Flow-assisted alignment in microchannels	(1) parallel and crossed NW arrays can be assembled; (2) compatible with both rigid and flexible substrates.	(1) area for NW assembly is limited by the size of fluidic microchannels; (2) difficult to achieve a very high density of NW arrays; (3) NW suspension needs to be prepared first.
Bubble-blown technique	(1) area for NW assembly is large; (2) compatible with both rigid and flexible substrates.	(1) it is difficult to achieve high-density NW arrays; (2) NW suspension needs to be prepared first.
Contact printing	(1) area for NW assembly is large; (2) high-density NW arrays can be achieved; (3) parallel and crossed NW arrays can be assembled; (4) direct transfer of NW from the growth substrate to the receiver substrate; (5) compatible with both rigid and flexible substrates; (6) NW assembly process is fast.	(1) the growth substrate needs to be planar; (2) the process works best for long NWs.
Differential roll printing	(1) area for NW assembly is large; (2) high-density NW arrays can be achieved; (3) direct transfer of the NW from the growth substrate to the receiver substrate; (4) compatible with both rigid and flexible substrates; (5) NW assembly process is fast.	(1) the growth substrate needs to be cylindrical; (2) the process works best for long NWs.
Langmuir-Blodgett technique	(1) area for NW assembly is large; (2) high-density NW arrays can be achieved; (3) parallel and crossed NW arrays can be assembled; (4) compatible with both rigid and flexible substrates.	(1) NWs typically need to be functionalized with the surfactant; (2) the assembly process is slow and has to be carefully controlled; (3) NW suspension needs to be prepared first.
Electric field-assisted orientation	(1) NWs can be placed at a specific location; (2) compatible with both rigid and flexible substrates; (3) NW assembly process is fast.	(1) patterned electrode arrays are needed; (2) area for NW assembly is limited by the electrode patterning; (3) NW density is limited; (4) it works the best for conductive NWs; (5) NW suspension needs to be prepared first.

*Source:* Reproduced with permission from Liu et al. [267]. Copyright 2012: American Chemical Society.

**Table 4 nanomaterials-10-01392-t004:** Parameters of gas sensors based on 2D metal oxide structures.

Nanosheets (Technology)	DT, nm	Response ^a),b)^	DL, ppm	Target Gas	τ_res_/τ_rec_ s	T_oper_, °C	Ref.
ZnO (ST)	30	~11^a)^@100 ppm	5.0	CO	25/36	300	[63]
P ZnO (HT)	~100	~84^a)^@50 ppm	1.0	C_2_H_5_OH	15/12	330	[200]
MP ZnO (HT)	~18	101^a)^@100 ppm	1.0	C_2_H_2_	11/5	400	[364]
ZnO (FP)	~20	106^a)^@200 ppm	<1.0	CH_3_COCH_3_	19/14	300	[374]
ZnO (SC)	10–60	~75%^b)^@1 ppm	0.05	HCHO, CH_3_CHO	10/62	220	[375]
ZnO:Sn-Rh (HT)	-	15^a)^@100 ppm	5.0	C_2_H_5_OH	3/10	300	[378]
SnO_2_ (HT)	~15	40^a)^@100 ppm	5.0	C_2_H_5_OH	1/9	300	[366]
SnO_2_ (HT)	~20	~70^a)^@100 ppm	100	CO	9/18	300	[367]
SnO_2_ (HT)	~10	~70^a)^@100 ppm	<5.0	C_2_H_5_OH	-	250	[382]
		~10^a)^@500 ppm	<50	CO	1/3	300	
WO_3_ (HT)	~10	~6^a)^@50 ppb	0.05	NO_2_	140/75	140	[368]
WO_3_ (HTA)	10–50	80%^b)^@ ppm	600	H_2_	120/235	250	[383]
WO_3_/SnO_2_ (HT)	30–90	45^a)^@100 ppm	~1.0	C_3_H_6_O	2/30–60	260	[379]
CuO (FHT)	~19	~9^a)^@100 ppm	10	C_2_H_5_OH	15/11	370	[369]
CuO (FHT)	~63	~10^a)^@100 ppb	<0.01	H_2_S	234/76	RT	[384]
P NiO (HT)	~14	13^a)^@10 ppm	0.001	H_2_S	100/79	92	[385]
NiO (ACD)	~100	~650%^b)^@8 ppm	0.008	HCHO	120/120	150	[386]
Co_3_O_4_ (FHT)	20–50	~14^a)^@100 ppm	10	C_2_H_5_OH	N/A	160	[387]
Co_3_O_4_ (FHT)	~40	9^a)^@100 ppm	0.2	NH_3_	9/134	RT	[371]
In_2_O_3_ (FTSS)	~3.5	213^a)^@10 ppm	0.01	NO_x_	4/9	120	[372]
In_2_O_3/_WO_3_ (4wt%) (IM)	~2.5	25^a)^@100 ppm	0.1	HCHO	1/67	170	[380]
V_2_O_5_·0.76H_2_O (MOA)	1.5–2.6	2%^b)^@10 ppm	10	H_2_	–	250	[373]
RuO_2_ (LPE)	~1.3	1.1%^b)^@20 ppm	5	NO_2_	–	80	[388]

ACD—aqueous chemical deposition process; DL—detection limit; DT—thickness; FP—facile precipitation method; FHT—facile hydrothermal process; HTA—high temperature anodization; FTSS—facile two-step synthetic method; HTS – hydrothermal synthesis; IM—impregnating method; MOA—molecular-level oriented attachment method; MP—mesoporous; LPE—liquid phase exfoliation technique; P—porous; Response: ^a)^ R_air_/R_gas_; ^b)^ ΔR/R_0_ (%); RT—room temperature; SC—sonochemical method; ST—solvothermal process; T_oper_—operation temperature; τ_rec_—recovery time; τ_res_—response time.

**Table 5 nanomaterials-10-01392-t005:** Comparison between the different deterministic placement methods. Qualitative comparison in terms of cleanness, easiness, and speed between the different deterministic placement methods described in the text. Comments about their main drawbacks are also included in the table.

Method	Cleanness	Easiness	Speed	Notes
PMMA carrier layer	***	***	***	Spin-coating is needed, direct contact with the polymer, it can transfer large-area flakes.
Elvacite sacrificial layer	*	***	***	Capillary forces, spin-coating is needed, direct contact with the polymer.
Wedging	*	**	***	Capillary forces, dip-coating is needed, difficult alignment, direct contact with the polymer, transfer over curved or uneven surfaces is possible.
PDMS dry transfer	***	*****	*****	Direct contact with the polymer.
Van der Waals pick-up	*****	*	**	Spin-coating is needed, several steps involved, only works to transfer heterostructures, direct contact with the polymer only for the topmost layer.

*Source*. Reproduced with permission from Frisenda et al. [403]. Copyright 2018: Royal Society of Chemistry.

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
