# Peer review of "Current Trends in Nanomaterials for Metal Oxide-Based Conductometric Gas Sensors: Advantages and Limitations. Part 1: 1D and 2D Nanostructures"

_nanomaterials, 2020, doi:10.3390/nano10071392_

Round 1

Reviewer 1 Report

In the manuscript nanomaterials-844205, the authors have discusses the state-of-art advancements in 1D and 2D nanomaterials for conductometric gas sensors based on metal oxides. This is a nice piece of work and I suggest a ‘major revision’, however the authors need to answer the following concern.

  1. Abstract: “…using in the development of conductometric..”. Please correct the sentence by removing ‘in’.
  2. The ‘Abstract’ ending is not appropriate. It is currently demonstrating the limitations and disadvantages only. While the Title illustrates that both the advantages and limitation will be discussed, the abstract significantly failed to justify it. It should be strictly improved.
  3. “ In the literature you can find a huge number of works…”. The sentence should be improved. ‘You can find..’ is not a correct language in scientific articles. Besides it is also confusing that to whom the author is referring to by saying ‘you’. Please change this.
  4. “In this review, we will not follow this tradition and try to give a realistic..”. The ‘and’ should be replaced with ‘but’ or any similar word with correct grammar.
  5. “These sensors are easy-to-manufacture devices with simple operation..”. Remove the word ‘device’ here.
  6. “As you can see, for conventional gas sensors…” Remove the word ‘you’ here and anywhere throughout the text.
  7. The text is lacking the coherence of facts and statements. The introduction should be explicitly improved. Important functional sensing nanomaterial based review can be cited: https://doi.org/10.1063/1.5123479; https://doi.org/10.1021/cr400659h
  8. Besides, the author failed to address the uniqueness in this review and as such why this review is important for the gas sensing field and how it can address the loop holes currently present in sensing domain.
  9. Important and currently trending 2D materials (e.g. TMDs, graphitic nitrides, Mxenes etc) are missing from the introduction section and so can be improved by citing the more relevant literatures based on 2D materials e.g graphitic carbon nitride; https://doi.org/10.1039/C8TA02702A; https://doi.org/10.1039/C6NR08039A; https://doi.org/10.1039/C7TA02860A and TMDs https://doi.org/10.1007/s00604-018-2750-5

Author Response

R.1. In the manuscript nanomaterials-844205, the authors have discusses the state-of-art advancements in 1D and 2D nanomaterials for conductometric gas sensors based on metal oxides. This is a nice piece of work and I suggest a ‘major revision’, however the authors need to answer the following concern.

G.K. Thanks. However, your statement “This is a nice piece of work” is not consistent with the subsequent sentence “I suggest a ‘ major revision ’. Maybe you meant a ’minor revision’?

R.1. Abstract: “…using in the development of conductometric..”. Please correct the sentence by removing ‘in’.

G.K. For me it is O.K. However, according to your request I have modified the sentence.

This article discusses the main uses of 1D and 2D nanomaterials in the development of conductometric gas sensors based on metal oxides”

R.1. The ‘Abstract’ ending is not appropriate. It is currently demonstrating the limitations and disadvantages only. While the Title illustrates that both the advantages and limitation will be discussed, the abstract significantly failed to justify it. It should be strictly improved.

G.K. Dear reviewer 1, I disagree with you. The abstract clearly states that nanomaterials have advantages that can improve the parameters of sensors.

“…It is shown that, along with the advantages of these materials, which can improve the parameters of gas sensors, there are a number of disadvantages that significantly limit their use in the development of devices designed for the sensor market….”

I cannot list all advantages of nanomaterials. It is better to get acquainted with them in the process of reading the article. As you can see, I also did not list the shortcomings in the abstract

R.1. “ In the literature you can find a huge number of works…”. The sentence should be improved. ‘You can find..’ is not a correct language in scientific articles. Besides it is also confusing that to whom the author is referring to by saying ‘you’. Please change this.

G.K. You are right. I modified the sentence.

“In the literature one can find a large number of works aimed at promoting advantages of nanomaterials for various applications”

R.1. “In this review, we will not follow this tradition and try to give a realistic..”. The ‘and’ should be replaced with ‘but’ or any similar word with correct grammar.

G.K. O.K. I modified the sentence.

“..In this review, we will not follow this tradition. We will try to give a realistic look at the problem of using nanomaterials and show that, along with obvious advantages, 1D and 2D nanomaterials have disadvantages, which in some cases can significantly limit their application….”

R.1. “These sensors are easy-to-manufacture devices with simple operation..”. Remove the word ‘device’ here.

G.K. O.K. I modified the sentence.

“….These sensors are easy-to-manufacture and characterized by low production costs…”                                                                                                                                

R.1. “As you can see, for conventional gas sensors…” Remove the word ‘you’ here and anywhere throughout the text.

G.K. O.K. I modified the sentence.

“…Mentioned above testifies that for conventional gas sensors designed on the base on polycrystalline metal oxides, there are quite specific requirements for parameters of gas-sensitive materials….”

G.K.  I also removed “you” in other sentences.

R.1. The text is lacking the coherence of facts and statements. The introduction should be explicitly improved. Important functional sensing nanomaterial based review can be cited: https://doi.org/10.1063/1.5123479; https://doi.org/10.1021/cr400659h

G.K. I did not understand this remark. What coordination of facts and statements are you talking about?

The introduction accurately determines the purpose of this review and justifies the selection of objects for analysis. The main statements and their justification are made in the following sections of the review.

G.K. O.K. I will include your article (https://doi.org/10.1063/1.5123479) in the reference list. It will be [78].

In addition, the article https://doi.org/10.1021/cr400659h discusses polymer-based composites. Unfortunately, this issue is not considered in this article. Therefore, this article cannot be included in the list of references. If I include all the articles related to metal oxide nanomaterials, then my list of references will include more than 1000 links.

R.1. Besides, the author failed to address the uniqueness in this review and as such why this review is important for the gas sensing field and how it can address the loop holes currently present in sensing domain.

G.K. It is strange that you did not understand the uniqueness of this review. This could be understood already in the first paragraph of the article.

«…We will try to give a realistic look at the problem of nanomaterials using and show that, along with obvious advantages, 1D and 2D nanomaterials have shortcomings, which in some cases can significantly limit their application….»

I also did not understand what holes I should close? In my reviews and books, I already tried to close so many holes currently present in gas sensing domain that I was already confused. Don’t you really understand the main message of the article - nanomaterials cannot solve all the problems of gas sensors. And therefore, it is necessary to look realistically at the prospects for their use in gas sensors designed for the market.

R.1. Important and currently trending 2D materials (e.g. TMDs, graphitic nitrides, Mxenes etc) are missing from the introduction section and so can be improved by citing the more relevant literatures based on 2D materials e.g graphitic carbon nitride; https://doi.org/10.1039/C8TA02702A;  https://doi.org/10.1039/C6NR08039Ahttps://doi.org/10.1039/C7TA02860A and TMDs https://doi.org/10.1007/s00604-018-2750-5

G.K. Dear reviewer. I clearly wrote that “... The most famous 2D nanomaterials are graphene [8, 323], black phosphorus or phosphorene [324], silicene, borophene, boron nitride [325], and transition metal dichalcogenides (TMDs) [326], which have a layered structure and allow the formation of atomically thin 2D nanomaterials of a large area. " Graphitic carbon nitride does not apply to the most famous 2D nanomaterials and therefore I will not include your articles in the list of references. You must understand that I cannot list all 2D materials and give numerous references to them.

Reviewer 2 Report

The review was written by Ghenadii Korotcenkov, who is a well-known specialist in the field of gas sensors, the author of a large number of reviews, and the editor of many books in the field of metal oxides and gas sensors.

The review is devoted to materials for conductometric gas sensors based on nanocrystalline metal oxides of 1D and 2D dimensions. Very different aspects of sensing materials such as: synthesis methods and techniques, chip engineering, sensor performances, sensor market are considered. The review is positively distinguished by a good list of cited literature, which includes the main significant works in this area. The review is designed in such a way that the problems of dimensionality and morphology of sensor materials are key, which is certainly of great interest to specialists and corresponds to the scope of Nanomaterials. The author's conclusions about the advantages and disadvantages of 1D and 2D materials are mostly correct.

Considering the review as a whole, I note that, despite the large volume of the article, the author was unable to find a consistent approach: physical (stress, band bending), structural (reconstruction of the surface) or chemical (electronic state of atoms, metal-oxygen bond energy) for discussion the role of dimension on the sensor properties of metal oxides. A large number of reviews and books have been written on this subject, especially 1D metal oxides, without understanding the role of dimensionality on processes taking place on the surface, this looks like a simple listing of existing results. The complexity of this approach lies in the variety of metal oxides such as ZnO, In2O3, CuO, Ga2O3, MnO2, CeO2, TiO2, WO3, which are characterized by a completely different surface structure and, as a consequence, by a different nature and concentration of active centers on the surface. Different synthesis conditions, raw materials, crystallization temperatures even for one material lead to a change in the surface structure and make it difficult to discuss the impact of dimensionality on sensor performances.

Given the above, I believe that the review can be published after major revision of content and reduction in volume. It would be useful to reduce the number of oxides and show, with a few examples ZnO, TiO2, SnO2 the role of dimension in sensor behavior. Descriptions and diagrams of “top down” and “bottom up”  techniques for the synthesis of nanomaterials are given in different textbooks on nanotechnology and can be removed.

Author Response

R.2. The review was written by Ghenadii Korotcenkov, who is a well-known specialist in the field of gas sensors, the author of a large number of reviews, and the editor of many books in the field of metal oxides and gas sensors.

The review is devoted to materials for conductometric gas sensors based on nanocrystalline metal oxides of 1D and 2D dimensions. Very different aspects of sensing materials such as: synthesis methods and techniques, chip engineering, sensor performances, sensor market are considered. The review is positively distinguished by a good list of cited literature, which includes the main significant works in this area. The review is designed in such a way that the problems of dimensionality and morphology of sensor materials are key, which is certainly of great interest to specialists and corresponds to the scope of Nanomaterials. The author's conclusions about the advantages and disadvantages of 1D and 2D materials are mostly correct.

G.K. Thanks

R.2. Considering the review as a whole, I note that, despite the large volume of the article, the author was unable to find a consistent approach: physical (stress, band bending), structural (reconstruction of the surface) or chemical (electronic state of atoms, metal-oxygen bond energy) for discussion the role of dimension on the sensor properties of metal oxides. A large number of reviews and books have been written on this subject, especially 1D metal oxides, without understanding the role of dimensionality on processes taking place on the surface, this looks like a simple listing of existing results. The complexity of this approach lies in the variety of metal oxides such as ZnO, In2O3, CuO, Ga2O3, MnO2, CeO2, TiO2, WO3, which are characterized by a completely different surface structure and, as a consequence, by a different nature and concentration of active centers on the surface. Different synthesis conditions, raw materials, crystallization temperatures even for one material lead to a change in the surface structure and make it difficult to discuss the impact of dimensionality on sensor performances.

G.K. I agree with the reviewer that this review does not address the influence of the dimension effect on the sensor properties of metal oxides in the broadest sense of the word, namely the effect of stress, band bending, reconstruction of the surface, electronic state of atoms, and metal-oxygen bond energy on the sensing properties of metal oxides. However, I would like to note that such consideration was not the purpose of this article. This is clear from the title of the article. To consider this problem, a completely different approach is required, which should be based on the results of peculiar studies combined with theoretical calculations. It is important to note that this problem is common to all metal oxide materials and requires separate consideration. I would be happy if the reviewer shows me a review, where an attempt was made to consider and analyze the proposed aspects of the influence of dimension on the sensor properties of metal oxides.

At the same time, I do not agree that my review looks like a simple listing of existing results. If you compare my review with most reviews related to metal oxide sensors, you will see that my review is not a simple listing of the results obtained by various groups. Unlike many authors, I try to generalize and understand the general laws characteristic of this group of materials.

R.2. Given the above, I believe that the review can be published after major revision of content and reduction in volume. It would be useful to reduce the number of oxides and show, with a few examples ZnO, TiO2, SnO2 the role of dimension in sensor behavior.

G.K. I must say that I tried to present information in this article very briefly, and therefore I can’t imagine how it is possible to reduce the volume of the article. Regarding the role of the size of nanomaterials in the behavior of the sensor, I should note that some aspects of the influence of the size of nanowires and 2D nanomaterials on the sensor response are considered in this article. In addition, I must say that this topic is also of interest to me (for example, see my reviews: G. Korotcenkov, B.K. Cho, The role of grain size on the thermal stability of nanostructured metal oxides used in gas sensor applications and approaches for grain-size stabilization, Prog. Crystal. Growth 58 (2012) 167-208; G. Korotcenkov, S.D. Han, B.K. Cho, V. Brinzari, Grain size effects in sensor response of nanostructured SnO2- and In2O3-based conductometric gas sensor, Crit. Rev. Sol. St. Mater. Sci. 34 (1-2) (2009) 1-17, G. Korotcenkov, The role of morphology and crystallographic structure of metal oxides in response of conductometric-type gas sensors, Mater. Sci. Eng. R. 61 (2008) 1-39, and G. Korotcenkov, Gas response control through structural and chemical modification of metal oxides: State of the art and approaches, Sens. Actuators B 107(1) (2005) 209-232). Therefore, it is possible that in subsequent reviews I will consider this problem. But not in this review, where a completely different task was posed. It is important to note that for such a consideration it is absolutely not necessary to reduce the number of metal oxides to be analyzed.

R.2. Descriptions and diagrams of “top down” and “bottom up” techniques for the synthesis of nanomaterials are given in different textbooks on nanotechnology and can be removed.

G.K. If we will start to use your approach to review articles, then only the title should be left from the review articles, since most of the topics discussed in them have already been described in one form or another in different textbooks. If we remove the description of “top down” and “bottom up” techniques for the synthesis of nanomaterials from this review, we will reduce the article size by only 1.5 pages of text. Such savings due to loss of integrity of the presentation of the material in question is not rational from my point of view.

Round 2

Reviewer 1 Report

The author have answered to most (not all) of my queries in an interesting way and so my decision is manuscript 'accept'. 

Reviewer 2 Report

The review is devoted to materials for conductometric gas sensors based on nanocrystalline metal oxides of 1D and 2D dimensions. Very different aspects of sensing materials such as: synthesis methods and techniques, chip engineering, sensor performances, sensor market are considered. The review is positively distinguished by a good list of cited literature, which includes the main significant works in this area. The review is designed in such a way that the problems of dimensionality and morphology of sensor materials are key, which is certainly of great interest to specialists and corresponds to the scope of Nanomaterials. The author's conclusions about the advantages and disadvantages of 1D and 2D materials are mostly correct.

Accept in present form